# Neural Attention Search

**Difan Deng**
Leibniz University Hannover
d.deng@ai.uni-hannover.de

**Marius Lindauer**
Leibniz University Hannover
L3S Research Center
m.lindauer@ai.uni-hannover.de

## Abstract

We present Neural Attention Search (NAtS), an end-to-end learnable sparse transformer that automatically evaluates the importance of each token within a sequence and determines if the corresponding token can be dropped after several steps. To this end, we design a search space that contains three token types: (i) Global Tokens will be preserved and queried by all the following tokens; (ii) Local Tokens survive until the next global token appears; and (iii) Sliding Window Tokens have an impact on the inference of a fixed size of the next following tokens. Similar to the One-Shot Neural Architecture Search approach, this token-type information can be learned jointly with the architecture weights via a learnable attention mask. Experiments on both training a new transformer from scratch and fine-tuning existing large language models show that NAtS can efficiently reduce the KV cache size and the inference costs for the models while maintaining the models' performance.

## 1 Introduction

The ability to understand and infer from long-context information is crucial for many tasks such as long document summarization [94] and question answering [20, 48], code generation [35, 56] or multi-round dialogues [91]. Thanks to the ability to query the information from any position of the historical sequence, transformer-based large language models [11, 14, 32, 34, 42] extend their context length up to millions of tokens.

However, querying information from historical sequences requires a complexity of $\mathcal{O}(L^2)$ w.r.t. the input sequence length $L$. KV caching could reduce this time complexity to $\mathcal{O}(L)$ by storing all the historical KV values. Nevertheless, with the increasing model size of recent LLMs, even the $\mathcal{O}(L)$ time-wise and memory-wise complexity could become a bottleneck during inference time.

Indeed, not all the tokens in a sequence are equally important [46]. Many of them are redundant and do not contribute to the final output. Humans can recognize this information without pre-defined fixed rules and summarize or discard the context information into much smaller content. Transformers could also learn this ability implicitly: Many tokens in the attention map might only have very low weights [96] and only have little influence on the final predictions. However, as the transformer learns this information implicitly, we might not know how the important tokens would be distributed in the context. Selecting these tokens and recognizing the attention distributions might require extra human experts' knowledge by either looking at the attention maps [27, 58, 95, 96] or applying specific fixed rules [13, 16, 17, 31, 84]. Since this knowledge is already contained in the transformer models, we could also ask the model to evaluate the importance of each token and learn to predict the optimal type for the given input tokens automatically.

Unlike prior works that rely on human expertise or predefined rules to identify important tokens [15, 27, 28, 31, 54, 84, 85, 96], we propose a novel approach to evaluate the importance of each token by assigning different roles to each of the tokens. For example, some tokens will be preserved until the end, while other tokens might only survive for a short amount of time. These roles measure

39th Conference on Neural Information Processing Systems (NeurIPS 2025).

the importance of each token and determine if it would survive within the next few tokens. Rather than pre-defining a set of fixed rules for each token, we ask the model to learn this information automatically. Finding the optimal role for each token is similar to the process of neural architecture search [23, 81], where an optimal architecture is searched by the optimizer for a given task. Thereby, in this work, we jointly optimize the choice of each token and the model weights by constructing a learnable attention mask. Our approach is implicitly motivated by the one-shot neural architecture search approach [22, 55, 68], where the model parameters and architecture parameters are jointly optimized during the search process. However, our approach, Neural Attention Search (NAtS), searches for the optimal token roles jointly with the attention weights.

Our contributions are as follows:

1. We propose Neural Attention Search (NAtS), an end-to-end learnable sparse attention framework that automatically searches for the optimal roles for each token in the input sequence.
2. We introduce different token roles in our search space that can be later combined to construct a learnable attention mask and then jointly optimized with the model weights in NAtS.
3. We show that NAtS could efficiently reduce the KV cache required during inference time while maintaining most of the models' performance.

By automatically learning to focus on the most relevant information, NAtS paves the way for more efficient and scalable inference with LLMs in long-context applications.

## 2 Background and Related Work

### 2.1 Attention maps for transformers

Transformers [79] computes the correlation between different tokens by mapping the input sequences into three variables, $Q$, $K$, and $V$. An attention map $A$ is first computed by pairing $Q$ and $K$ and then scaled and normalized with a softmax function. Finally, this attention map $A$ is multiplied by $V$. Additionally, a mask $M$ is attached to the Attention Map to guide the attention to focus only on certain tokens. This can be expressed as

$$A = \frac{QK^T}{\sqrt{d_{head}}}, \qquad O = \text{softmax}(A + M^{add})V \qquad (1)$$

The additive attention mask $M_{i,j}^{add} \in \{-\inf, 0\}$ controls if the transformer needs to construct the correlation between $Q_i$ and $K_j$. A widely used mask $M^{add}$ used in transformers is a causal mask [79], where the upper parts of the mask are filled with $-\inf$; each token can only query the token information before it.

Computing the attention map of a sequence with length $L$ requires a complexity of $\mathcal{O}(L^2)$ for both time and memory. This complexity can be lowered to $\mathcal{O}(L_{\text{cache}})$ during inference time, with $L_{\text{cache}}$ being the length of the KV cache. However, as an expense, the cost of storing the KV cache gradually becomes unnegligible with the increasing size of large language models and the context length they could handle. Therefore, lots of work has been proposed to reduce the storage and computation overheads during the inference time [52].

To reduce the computational costs of an LLM on long context information, LLMLingua [66] trains another smaller model to compress the input texts. Many other studies work on reducing the KV cache size by deciding which tokens to evict or preserve with the past attention maps. Recent works such as H2O [96], Scissorhand [58], FastGen [31], KeyFormer [6], SnapKV [54], PyramidKV [15], AadaKV [27], ChunkKV [57], and CriticalKV [28] all apply different rules to identify the importance of each token and remove all the remaining tokens. Additionally, instead of simply removing the unimportant tokens, we can also merge them into the existing tokens [80, 95].

Other works do not aim at reducing the KV cache size, but only select a subset of KV caches to compute the attention outputs, including Quest [76], TokenButler [9], AttentionPredictor [87], and XAttention [86]. However, these works still need to maintain the entire KV cache in the GPUs. This still results in a huge GPU memory usage overhead. To alleviate this issue, some other works like InfLLM [83], HIP [51], and ShodwKV [75] offload the KV cache to CPUs and only load the KV caches that are relevant for the current token predictions to GPUs.

While these existing methods offer various ways to reduce KV cache size, they often rely on inflexible predefined rules and potentially inaccurate heuristics based on past attention maps. NAtS, in contrast, introduces a novel approach that learns to dynamically assign token roles during inference, enabling a more adaptive and efficient use of the KV cache. By treating token role assignment as an optimization problem, NAtS leverages principles from neural architecture search to jointly optimize token roles and model weights, leading to a more flexible and powerful attention mechanism.

## 2.2 Sparse Attention

Sparse attention works on computing only a fraction of the attention map to reduce the computation costs and KV cache sizes with either pre-defined fixed patterns, including Sparse Transformers [17], Longformer [13], StreamingLLM [84], and DuoAttention [85]. Alternatively, this can be adjusted by special embedding tokens such as Longcoder [35] and SepLLM [16]. Additionally, one can also determine the types of sparse attention with a set of proxy values, including MInference [43], FlexPrefill [50], SeerAttention [30], SpargeAttention [93], and XAttention [86]. However, these approaches aim to approximate the attention output of each layer from a pre-trained dense transformer that only involves the QK matrices. In contrast, NAtS learns the token roles directly from the final loss function and does not necessarily need to approximate the existing attention output. Hence, NAtS can be applied to both training from scratch or approximating the outputs from an existing model. Additionally, the existing approaches can only be applied to approximating the attention maps during the pre-filling stages and still need to preserve the entire KV cache during the decoding stage, making them less profitable for reasoning models where the model needs to generate lots of tokens during the decoding stage [21]. In contrast, NAtS only checks the roles for each token and does not necessarily need to link to the actual attention map values. Hence, it can be applied to both pre-filling and decoding stages while keeping the KV cache sizes to a low level.

In contrast to the fine-tuning-free approaches above, we can also fine-tune the target model to achieve the desired sparsity. This includes Adaptively Sparse Attention [10] and Dynamic Memory Compression [64]. However, both approaches rely on expensive cumulative productions during training time, making their approach less applicable to long-context scenarios. LLMLingua2 [66] trains another model to compress the target texts. Additionally, Kim et al. [46] asks the model to learn to drop the tokens with lower attention scores. Landmark Attention [63] inserts landmark tokens after certain time steps and applies these landmark tokens to summarize the information of the tokens before that landmark token and recover. SPARSEK [60] only selects a constant number of KV pairs for each query by introducing a differentiable SparseK operator. Mixture-of-Depth [72] only passes a limited number of tokens to each transformer layer. Native Sparse Attention [92] and MoBA [61] all select a subset of the KV values to compute the attention masks and therefore reduce the computational overhead. However, these approaches still require preserving the entire KV cache values and only reducing computational costs during decoding stages.

MoA [29] proposes to search for the optimal sparse attention for different heads, which share similar ideas to NAtS. However, MoA's search space only contains streaming LLM. This head-level search space of MoA is much smaller than our token-level search space, which might restrict the model's expressibility under even smaller budgets.

## 2.3 Neural Architecture Search

Designing a neural architecture for a specific task might require a lot of trial and error. Neural Architecture Search (NAS) automates this process by searching in a pre-defined search space [24]. Previous work on NAS mainly focused on searching within a discrete search space by sampling a new architecture from the search space, evaluating its performance, and updating the optimizers [98, 99]. One-Shot NAS [68] approaches instead share the same weights of the operators with respect to all the architectures in the search space. This allows to jointly optimize the architectures and weights. DARTS [55] and GDAS [22] further relax the discrete search space into continuous values to optimize the architecture parameters with gradient descent. The One-Shot NAS approach allows the optimizers to efficiently search for the optimal architecture within a huge search space. Similarly, NAtS has multiple options for each token as the search space and is able to search for the optimal token types jointly with the model weights. However, unlike One-Shot NAS approaches that consider the optimization problem as a bilevel optimization problem and optimize the model weights and architecture weights alternatively, we optimize the token state information and model weights within

one forward and backward pass. This is similar to a mixture-of-experts (MOE) model [26, 73]. However, instead of distributing data across all experts uniformly, we only select one expert for each token and assign all data to that expert.

# 3 The NAtS Approach

In this section, we first introduce all the candidate token types in our search space. We then show that we can construct a learnable attention mask with the choice of each token type. Finally, we can efficiently reduce the KV cache size by dropping the unnecessary tokens during inference.

## 3.1 Search Space Design

Not all the tokens in a sequence are equally important. Just like a paragraph is composed of multiple sentences, a sequence can be divided into multiple sub-sequences containing different information; some tokens might only be required within these sub-sequences. Hence, we design a search space for each token's role within the sequence and ask the model to automatically learn the optimal role for each token in the sequence.

We first define **Global Tokens** as tokens containing important information that need to be preserved for the following predictions. Liu et al. [58] and Zhang et al. [96] showed that only a small fraction of the tokens contribute to most of the attention scores for self-attention computation. These tokens need to be preserved for models to recall the global information that helps to predict the next token. In vanilla transformer models, all the tokens are Global Tokens.

Global Tokens will not be evicted during inference time. Therefore, we should maintain as few Global Tokens as possible to ensure inference efficiency. Each Global Token should not only preserve the information within that position. Ideally, it should also be able to summarize the information from previous sequences [16, 35]. Hence, we split the entire sequence with the Global Tokens into multiple sub-sequences, with each sub-sequence ending with one Global Token. Each Global Token only needs to summarize the information from its sub-sequences and the previous Global Tokens.

**Local Tokens** only survive until the next Global Token appears. Therefore, models will have the full attention within each sub-sequence to summarize the local sub-sequence information into the Global Token which is located at the end of the sub-sequence. Meanwhile, the model will be sparse within the input sequence. Local Tokens are considered as tokens that only provide lower-level information that helps the model to understand the local context information, but might not help further outside this context. This provides the model with a flexible interface to control its sparsity based on the input context information.

Only the Global Tokens and Local Tokens might control the sparsity at a low granularity level. E.g., assuming that one input sequence is highly localized, each token only has a high correlation with itself or a few neighboring tokens. In this case, they are all similar and are assigned with the same token type. However, none of the Global Token and Local Token could sparsify this attention map efficiently: if all the tokens are classified as Local Tokens, the input sequence will only be considered as one single subsequence, and all the Local Tokens will be equivalent to the Global Tokens.

Hence, we introduce **Sliding Window Token**. Sliding Window Tokens will only be preserved for the next $W$ time steps and were previously considered as one of the most popular sparse attention approaches [17, 31, 84, 96].

In contrast to other causal attention maps, Figure 1(d) illustrates an exemplary attention mask constructed by the choices of different token types. In this case, we define the sliding window size as 4. Token 1, 4, 10 act as Global Tokens; Tokens 2, 6, 7, 8, 11 are Local Tokens; Token 3, 5, 9, 11 are Sliding Window Tokens. The Global Tokens splits the entire sequence into three subsequences that end at 4, 10, and the last index, respectively. Hence, Token 2 will only be required by Token 3 and 4. This rule applies the same for Token 6, 7, and 8, where they only interact with the tokens within the same subsequence. For the sliding window tokens, only the next three tokens could query their information, regardless of whether these tokens belong to the same sub-sequence. Only 6 out of 12 tokens are involved during inference time to make the next token prediction.

Combining different token types could already cover most of the sparse attention variables. For instance, assigning Global Tokens to the first few tokens and Sliding Window Tokens to all the other

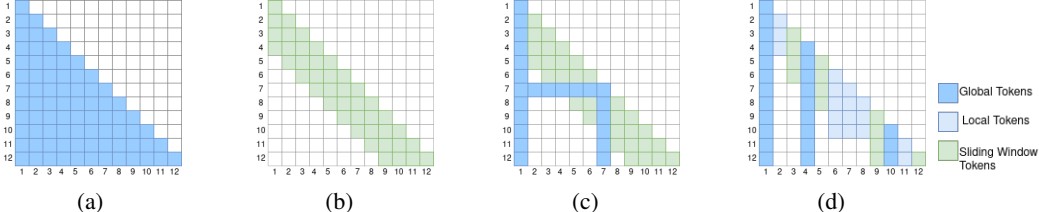

Figure 1: A comparison between different casual attention maps. 1(a): The full attention map, where each token is connected to the tokens before it. 1(b): the local attention with sliding windows 3, every token will only get access to the information of the 3 tokens ahead. 1(c) Longformer, besides the local attention, the first, 6th and 9th tokens are the pre-defined global tokens. 1(d): NAtS dynamically learns the optimal role for each token and constructs a learnable mask based on the tokens' roles.

tokens results in streaming LLMs [84] and MoA [29]. If we further set all the tokens in some heads as Global Tokens, we will get DuoAttention [83]. Assigning different tokens as Global Tokens results in the Vertical patterns [43] that appear in many Token eviction strategies [54, 58]. Since different layers or heads might have different numbers of global tokens, patterns like pyramidkv [15] and adakv [27] can be easily derived from our search space. For a transformer network with $N$ layers and $H$ attention heads that needs to process one sequence of length $L$, given the three token types defined above, there are $3^{L \times H \times N}$ possible configurations in our search space. It is prohibitive to check all the combinations in such a huge search space, with the increasing sequence length that a model receives. Here, we propose an end-to-end approach to search for the optimal token types together with the network weights jointly.

## 3.2 Searching for the Optimal Token Types

Searching for the optimal token types within a sequence is similar to searching for the optimal architectures for a given task in neural architecture search [24, 81]. Following GDAS [22], we apply the Gumbel-Softmax trick [41, 62] to sample from the discrete search space. The Gumbel-Softmax trick allows the gradient to be backpropagated through the discrete choice of the token types.

Specifically, we first use a linear layer (which we call Attention Score Layer) that maps each the input tensor for an attention layer $X \in \mathbb{R}^d$ to the likelihood for each option: $\alpha \in \mathbb{R}^{(H * N_{\mathrm{opts}})} = \mathrm{Linear}(\mathbf{x})$, where $H$ is the number of KV heads and $N_{\mathrm{opts}}$ is the number of options in the search space. The type $\alpha$ for each token is then sampled from a Gumble-Softmax function based on the likelihood values. Given that the number of options $N_{\mathrm{opts}}$ is normally much smaller than the model head dimensions, this linear layer only introduces minimal overhead compared to the QKV linear layers.

We now construct a learnable attention mask $M$ with a series of sampled token states. However, the additive mask in Eq. 1 will take $-\inf$ values, resulting in invalid gradient information. Hence, we use the token information to construct a multiplicative attention mask $M^{\mathrm{mul}} \in \{0, 1\}$ [71][1]:

$$O = \frac{e^A \odot M^{\mathrm{mul}}}{\sum_j e^{A_{.,j}} \odot M^{\mathrm{mul}}_{.,j}} V \qquad (2)$$

The attention mask columns for Global Tokens and Sliding Window Tokens can be directly constructed since they will survive for a fixed number of steps. However, the mask for Local Tokens $M^L_{i,j}$ is controlled by both the distribution from Local Tokens and Global Tokens as Local Tokens will survive until the next Global Token appears. In other words, to make $M^L_{i,j}(j > i)$ a valid value, no Global Token should appear between $i$ and $j$.

---

[1]For the sake of brevity, we will use $M$ instead of $M^{\mathrm{mul}}$ in the following part of this paper.

Formally, the attention masks can be created as follows:

$$M_{i,j}^{G} = 1 \tag{3}$$

$$M_{i,j}^{SW} = \begin{cases} 1 & \text{if } j \leq i + W \\ 0 & \text{if } j > i + W \end{cases} \tag{4}$$

$$M_{i,j}^{L} = \prod_{n=j+1}^{i-1} (1 - G_n) \tag{5}$$

where $W$ is the sliding window size and $G_n$ a Global Token at Position $n$. We then construct the attention masks based on the type of each token. After that, we mask out the upper triangular part of the mask to ensure its causality.

In practice, we first collect the index of the next global token $I_i^{G_{\text{next}}} := \min(\{k | k >= i \wedge G_k = 1\})$ and rewrite Eq. 5 as:

$$M_{i,j}^{L} = \begin{cases} 1 & \text{if } j \leq I_i^{G_{\text{next}}} \\ 0 & \text{if } j > I_i^{G_{\text{next}}} \end{cases} \tag{6}$$

During the backward process, we collect the column-wise gradient values from our attention masks and apply these gradients to update the parameters from our Attention Score Layer. Different from other approaches that approximate the attention maps or attention outputs from an existing model [9, 29, 87], NAtS can be learned directly from the target labels, and thus, can be applied for both pre-training a new sparse transformer from scratch or fine-tuning an existing transformer. To control the attention map's sparsity, we introduce a small regularization value $\lambda$ that is directly applied to the gradient for Global Token and Local Tokens to encourage more Sliding Window Tokens. This regularization value $\lambda$ can be considered as a value similar to a soft threshold, pushing the tokens whose contributions in the attention maps outside the scope of the sliding window sizes with lower levels towards Sliding Window Tokens. Hence, we could efficiently reduce the corresponding overall computational costs.

We show how the backward gradients are computed and how the $\lambda$ values control the attention map sparsity in the appendix C.

These rules are then integrated in FlashAttention [18, 19] to avoid explicitly computing the attention masks during the forward pass. In addition to the transformer computation, we only need to collect the next Global Token indices $\mathbf{I}^{G_{\text{next}}}$ (with a complexity of $\mathcal{O}(N)$) and then mask out the attention map values with the masks defined above. At the same time, we also skip all the computation blocks that do not contain any valid values (i.e., all the items in the mask of that block are 0) during the online attention process. Further details can be found in the appendix D

### 3.3 Efficient Inference with Different Token Types

NAtS can be applied to both pre-filling and decoding stages. During the pre-filling stage, we first compute the attention values for all the global and local tokens. We then compute attention map values for the sliding window tokens separately to fully utilize the parallelism of GPU architectures.

During the decoding stage, we dynamically map the input feature maps to the corresponding token types and discard the tokens no longer required by the following tokens once a new token arrives. The Sliding Window Tokens only survive for a fixed number of time steps. We preserve a queue in the KV cache space that only stores the most recent $W$ tokens and mask out the non-Sliding Window Tokens: when new tokens come, we place them after the tail of the queue to replace the tokens older than $W$.

Similar to the vanilla transformer forward process, when new tokens arrive, we concatenate them with the existing KV caches, generating new masks and computing the attention output. After that, we start our post-update process: we first check the state of each token to decide if we can remove them or keep them in the cached values. Since different heads might disagree on the types of the same token, we record both the sizes for Global Tokens ($Size_G$) and Local Tokens ($Size_L$) for all the heads. New Sliding Window Tokens do not change these sizes since they will always be placed separately. However, when a new Global Token for any head arrives, we remove all the Local Tokens from the

corresponding heads and place the new Global Token right after the existing Global Tokens and then update our $Size_G$ and $Size_L$ accordingly. The same strategy is applied when new $LocalTokens$ arrive: we place them at the end of the Local Tokens and increase the number for Local Tokens. A detailed updating process can be found under Appendix E.

Figure 2 illustrates an exemplary case to update the KV caches in NAtS with a sliding window size of 3. The first two places store the most recent tokens and use a mask to mask out the non-Sliding Window Tokens (yellow tokens). Since Token 8 for Head 1 is Sliding Window Token and Global Token for Head 2. We first move both tokens to the beginning of our cache to replace the old one. After that, we drop Token 8 in Head 1 since it has already moved to the sliding window caches. Then, since Token 8 in Head 2 is a Global Token, we drop all local tokens

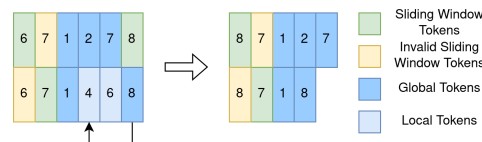

Figure 2: An example of how caches are updated within NAtS when new tokens arrive with a model containing two heads. The two rows represent different heads.

after the last Global Token (Token 1). Hence, we place Token 8 after Token 1 and remove Tokens 4 and 6. The $Size_G$ is then updated from $[5, 3]$ to $[5, 4]$ and the $Size_L$ is updated from $[0, 2]$ to $[0, 0]$: both new tokens are merged into the existing tokens, we do not need the extra space to store them.

## 4 Experiments

We implement NAtS based on the Flash Attention 2 [18] implementation on triton. In our experiments, we first train NAtS parameters jointly within a transformer model from scratch. We then apply NAtS to fine-tune a large language model to show that NAtS could efficiently reduce the KV cache size required while maintaining the model performance. The codes for NAtS implementation and experimental designs can be found under: `https://github.com/automl/NeuralAttentionSearch`

### 4.1 Training a Transformer From Scratch

We first apply NAtS to train a GPT2 small style [69] transformer model with 128M parameters from scratch on the PG-19 Language Modeling Benchmark [70] with four Nvidia H100-PCIe GPUs and evaluate it on the test sets of PG19 with a context length of 1024. Further details can be found under the appendix F.1.

As a baseline, we train another dense transformer model under the same hyperparameter setting. During inference time, we compare NAtS with the following baselines besides the full Transformer: (i) Streaming LLM [84] only preserves the first few starting and the most recent few tokens for future prediction. (ii) H2O [96] first computes the attention map and only preserves the tokens with the top-k attention scores. H2O and Streaming LLM are training-free approaches that control the sparsity with pre-defined hyperparameters during inference time. In contrast, NAtS controls this sparsity by the sparsity hyperparameter value $\lambda$. Hence, we train multiple models with different $\lambda$. However, in our experiments, we observe that the attention map sparsity values converge much faster than the model loss. We could quickly estimate the attention map sparsity within the first 10,000 iterations to check if this sparsity value satisfies the required sparsity and early-stop the runs that do not satisfy the requirements [40].

We provide different hyperparameters for H2O and Streaming LLM for different sparsity (with the sliding window size of 32, 64, 128, 256, plus the same number of HH for H2O and 64, 128, 256, 512, plus 8 Sink tokens for Streaming LLM). This results in a corresponding KV budget fraction ranging from 6.25% to 50%. Meanwhile, we train NAtS with the following $\lambda$: $0, 1e-9, 5e-9, 1e-8, 1e-7$. We run each experiment with three different seeds.

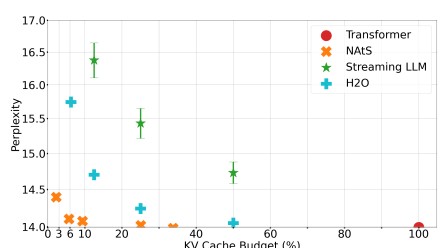

Figure 3: Perplexity vs KV Cache size under different sparsity settings $\lambda$ on the PG19 dataset.

Figure 3 shows the perplexity and the corresponding standard deviation (as error bars) of

the PG19 test sets from different model settings across different seeds. The x-axis indicates the fraction of KV caches applied to generate the last token in the input sequence. As expected, a larger $\lambda$ results in a smaller KV cache fraction, while the performance is relatively stable across different seeds. Interestingly, even if $\lambda$ is set as $0$, the KV cache budget can still be reduced to around 35% with slightly lower perplexity compared to the full attention. This might indicate that the model also tends to sparsify the input information to focus more on the relevant parts from the input data [89].

H2O maintains nearly lossless perplexity with a budget of 50%. However, as the available budget decreases, NAtS's perplexity remains nearly the same until a KV cache size of around 10%. In contrast, H2O and Streaming LLM start to degenerate their performance starting from a 25% budget allocation. The most sparse model in NAtS family only contains roughly 3% of the KV cache size (meaning roughly 30 cached tokens per layer on average) and achieves a lower perplexity compared to H2O with 12.5% of the KV caches.

## 4.2 Fine-Tune a Large Language Model

We now apply NAtS to fine-tune an existing large language model [34, 42] in the long context scenario. We construct a new training dataset that follows the construction rules of LongBench as a training set to fine-tune LLM models with NAtS. Some of LongBench's tasks are collected from the test sets of the previous benchmarks. Hence, we first collect the training sets from these benchmarks and construct these datasets following the data structure in LongBench. Additionally, we also add the passkey-retrieval dataset introduced in DuoAttention [83]. Overall, this dataset contains $7\,000$ instances with a maximal context length of $16\,000$. Further details can be found in Appendix F.2.

We only fine-tune the Attention Score Layer while keeping all the other parameters in the network fixed to approximate the original output from the corresponding base LLM. This approach is similar to DuoAttention [85]. However, since we want the model to capture the overall context information, we update Attention Score Layer with all the output from the full attention layer:

$$\mathcal{L}_{distill} = \frac{1}{B} \sum_{i=1}^{B} \sum_{j=1}^{L} (\mathbf{H}_{full}^{(i)}[j] - \mathbf{H}_{NAtS}^{(i)}[j])^2 \qquad (7)$$

where $B$ is the Batch Size. We only use the context information from the real-world dataset to ensure that the token information does not rely on specific prompts. However, the synthetic dataset might contain duplicated contexts, and hence, we only optimize the synthetic dataset with its corresponding labels [85]. Additionally, we control the sparsity with the regularization value $\lambda$ instead of the additional loss item defined in DuoAttention, and therefore, only optimize NAtS with $\mathcal{L}_{distill}$ as the loss function. We fine-tune two long-context models (Llama-3.1-8B-Instruct [34] and Mistral-7B-v0.3-Instruct [42]) on two Nvidia H100 PCIe GPUs for one epoch using AdamW [47, 59] with a learning rate of 2e-3 with a warm-up from 2e-4 in the first 20% steps and reducing back to 2e-4 in the last 20% steps. We apply different $\lambda$ to allow for different sparsity. For a fair comparison, we set the sliding window size $W$ as 256, the same as DuoAttention.

In addition to the baselines in Section 4.1, we also evaluate other state-of-the-art KV eviction strategies, including SnapKV [54], AdaKV [27], ChunkKV [57], PyramidKV [15], CriticalKV [28]. These approaches rely on a sliding window from the query matrix during the pre-filling stage to identify the important tokens. Unlike NAtS and the other approaches listed above that update the KV caches on the fly, these approaches only evict the KV cache once after the prefilling stage. Additionally, we evaluate DuoAttention [85] and MoA [29]. Both approaches extend the existing streaming LLMs approach to provide different budgets for different attention heads. These two methods share the same idea as NAtS, where the importance of different KV caches should be learned instead of the pre-defined rules. However, instead of learning the importance of the head level, we aim at learning this information directly on the token level.

Since we only optimize the Attention Score Layer, the number of learnable parameters is $n_{layers} \cdot d_{model} \cdot n_{heads} \cdot n_{options}$, where $n_{options}$ is the number of options in our search space (in our case, it is 3). Hence, the size of the parameters that need to be stored is negligible (i.e., it only takes roughly 13MB on disk for each set of Attention Score Layer) compared to the weights of the LLM. Hence, users could pre-collect all these weights and apply the one that fits their compression requirement.

We evaluate all the baseline results on the RULER [38] and LongBench [12]. RULER is a synthetic benchmark that evaluates a model's long-context capabilities across 4 task categories. LongBench

evaluates the model's capacity to understand information in different long context scenarios. Following the setting from KV Press [65], we ask all the one-time KV eviction approaches to evict the KV cache once the model receives all the context information before the question-related information arrives. For the other approaches, we evict the KV caches dynamically when new tokens are generated.

| | Full | Duo | SLLM | H2O | Snap | Ada | Chunk | Critical | Pyradmid | MoA | NAtS |
|---|---|---|---|---|---|---|---|---|---|---|---|
| 4k | 95.06 | 94.25 | 59.37 | 83.03 | 54.90 | 70.03 | 69.19 | 87.52 | 56.41 | 44.63 | **95.32** (51%) |
| 8k | 93.22 | 92.42 | 48.58 | 81.51 | 54.10 | 71.43 | 67.20 | 89.01 | 54.55 | 19.80 | **93.30** (45%) |
| 16k | 90.53 | 88.24 | 50.97 | 65.55 | 54.38 | 76.32 | 67.07 | 88.03 | 58.17 | 15.16 | **90.01** (42%) |
| 32k | 85.95 | 84.65 | 45.68 | 11.48 | 47.13 | 72.48 | 55.88 | 82.88 | 48.05 | 12.79 | **86.16** (41%) |
| 64k | 84.01 | 83.54 | 51.40 | 0.07 | 69.61 | 78.10 | 71.05 | 81.93 | 70.49 | 10.86 | **84.09** (41%) |
| 128k | 73.83 | 73.11 | 46.95 | 0.00 | 39.26 | 64.49 | 41.61 | 70.53 | 40.02 | 11.05 | **74.77** (44%) |

| | Full | Duo | SLLM | H2O | Snap | Ada | Chunk | Critical | Pyradmid | MoA | NAtS |
|---|---|---|---|---|---|---|---|---|---|---|---|
| 4k | 95.06 | 73.18 | 37.30 | 71.95 | 35.67 | 44.34 | 51.59 | 57.73 | 35.60 | 35.60 | **93.93** (24%) |
| 8k | 93.22 | 62.83 | 28.29 | 65.34 | 33.92 | 41.46 | 45.87 | 62.83 | 35.17 | 25.69 | **90.29** (19%) |
| 16k | 90.53 | 52.01 | 28.19 | 47.82 | 32.74 | 41.61 | 49.40 | 67.08 | 34.35 | 23.94 | **88.87** (16%) |
| 32k | 85.95 | 53.50 | 30.54 | 9.71 | 32.17 | 40.24 | 39.19 | 64.41 | 32.49 | 20.17 | **84.40** (15%) |
| 64k | 84.01 | 65.57 | 31.42 | 0.04 | 57.36 | 65.98 | 57.01 | 70.85 | 58.02 | 19.29 | **79.36** (15%) |
| 128k | 73.83 | 53.05 | 31.41 | 0.00 | 27.36 | 34.33 | 37.35 | 56.51 | 27.34 | 13.40 | **64.26** (16%) |

Table 1: Ruler results with 50% KV budgets (top) and 25% KV budget (bottom) on LLama 3.1. All Full models use the 100% KV budgets. We mark the actual KV budgets used by NAtS in the bracket. Models with the best performance besides the base Full Attention Models are bold

The results on the Ruler Benchmark with LLama 3.1 are shown in Table 1. NAtS uses $\lambda = 3e - 7$ for the 25% KV size level and $\lambda = 5e - 8$ for the 50% KV size level. NAtS achieves nearly lossless scores across different input context lengths. When the KV budget drops to 25%, NAtS achieves 96% of the full attention scores with much smaller KV budget sizes, while the other baselines could no longer keep their performance with the decreased KV budgets. Results with Mistral-7B-Instruct-v0.3 can be found under Appendix G.

For the LongBench tasks, we evaluate all the baselines with 50% and 25% of the KV cache sizes. We show the result with KV cache size of 25% for LLama3.1-8B in Table 2. The model that is used in this task is a model with $\lambda = 3e - 7$. NAtS achieves the best performance on most of the datasets with (in many cases) smaller KV cache values. We provide further results, including results with Mistral-7B and NAtS trained with other parameters, in Appendix G.

| | Full | Duo | SLLM | H2O | Snap | Ada | Chunk | Critical | Pyradmid | MoA | NAtS |
|---|---|---|---|---|---|---|---|---|---|---|---|
| NarrativeQA | 31.35 | 22.62 | 27.83 | 21.07 | 27.08 | 29.02 | 26.27 | 30.48 | 28.02 | 19.19 | **30.53**(13%) |
| Qasper | 24.73 | 18.99 | 14.53 | 15.00 | 14.24 | 14.84 | 12.71 | 15.79 | 13.43 | 18.57 | **23.76**(27%) |
| MultiFieldQA-en | 29.46 | 25.36 | 16.47 | 18.24 | 17.72 | 19.31 | 17.44 | 21.74 | 17.72 | 14.75 | **28.94**(24%) |
| MultiFieldQA-zh | 60.01 | 54.95 | 33.94 | 29.14 | 33.77 | 35.34 | 33.49 | 36.30 | 32.76 | 21.23 | **61.18**(22%) |
| HotpotQA | 17.06 | 16.06 | 13.95 | 16.47 | 15.58 | 16.12 | 16.50 | **16.73** | 14.23 | 10.18 | 16.06(20%) |
| 2WikiQA | 16.64 | 16.18 | 13.53 | 11.28 | 13.45 | 13.17 | 13.41 | 13.89 | 12.36 | 11.42 | **17.05**(23%) |
| Musique | 11.59 | 8.41 | 8.96 | 11.13 | 10.81 | 10.68 | 11.05 | 11.25 | 9.40 | 5.93 | **11.42**(18%) |
| DuReader (zh) | 35.56 | 32.18 | 18.64 | 24.77 | 23.57 | 24.57 | 24.24 | 24.68 | 23.56 | 22.33 | **34.28**(17%) |
| GovReport | 34.30 | 27.63 | 28.20 | 26.90 | 28.51 | 28.28 | 28.86 | 29.58 | 27.53 | 25.24 | **33.82**(19%) |
| QMSum | 23.30 | 22.28 | 20.67 | 20.39 | 21.41 | 22.05 | 21.27 | 22.27 | 21.98 | 19.59 | **23.11**(17%) |
| MultiNews | 27.13 | 24.58 | 21.76 | 22.80 | 23.14 | 23.50 | 22.72 | 24.03 | 23.00 | 23.30 | **26.72**(33%) |
| VCSUM (zh) | 16.36 | 14.97 | 14.26 | 14.85 | 14.80 | 15.18 | 14.92 | 15.46 | 15.21 | 14.58 | **15.61**(14%) |
| TREC | 72.50 | 58.50 | 66.00 | 55.00 | 53.50 | 59.00 | 54.50 | 62.00 | 48.50 | 60.50 | **72.00**(26%) |
| TriviaQA | 91.15 | 82.81 | 90.69 | 89.98 | 91.00 | 91.55 | 90.72 | 90.97 | 90.92 | 71.51 | **91.61**(22%) |
| SAMSum | 43.72 | 40.14 | 41.97 | 41.98 | 43.59 | 43.11 | 42.32 | 43.79 | 43.15 | 42.30 | **44.21**(16%) |
| LSHT | 46.50 | 35.00 | 35.50 | 27.50 | 45.00 | 44.00 | 43.50 | 46.00 | 45.00 | 22.00 | **47.50**(16%) |
| Passage Count | 6.63 | 4.50 | **7.22** | 3.40 | 5.52 | 5.14 | 4.11 | 5.90 | 4.70 | 3.79 | 7.12(20%) |
| PassageRetrieval-en | 97.98 | 91.65 | 94.81 | 72.25 | 90.36 | 90.70 | 90.38 | 94.70 | 88.32 | 28.21 | **95.81**(19%) |
| PassageRetrieval-zh | 77.99 | 55.22 | 28.97 | 52.13 | 73.06 | 78.74 | 55.95 | **80.15** | 74.66 | 24.78 | 78.5(20%) |
| LCC | 54.10 | 54.27 | 52.38 | 54.52 | **55.40** | 54.93 | 54.12 | 54.26 | 54.90 | 54.14 | 53.54(38%) |
| RepoBench-P | 51.39 | **57.26** | 49.65 | 55.54 | 51.80 | 52.44 | 52.87 | 52.57 | 52.41 | 53.35 | 53.48(28%) |

Table 2: LongBench Results with 25% Budget Allocation. The Full model uses the 100% KV budgets. The numbers in the brackets for NAtS are the used value KV cache sizes. We bold the models with the best performance besides the base Full Attention Models .

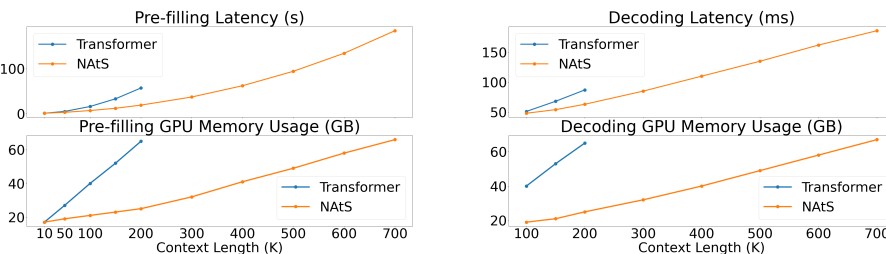

Figure 4: Memory and latency usage during pre-filling (left) and decoding (right)

## 4.3 Latency Evaluation

We evaluate the latency and memory usage with the full attention Llama 3.1 with NAtS (a $\lambda = 3e-7$) during both pre-filling and decoding phases with different context lengths. The experiments are implemented on a single Nvidia H100 PCIe GPU, and the model is stored as Bfloat16. We adapted RoPE [74] and RMSNorm kernel from FlashInfer [90] to accelerate the forward process. We apply chunked-prefilling [7, 49] with a chunk size of $10\,000$ by dividing the input contexts into multiple smaller chunks to reduce the peak memory usage from the intermediate activation values in the FFN layers. The full attention transformer quickly runs out of the GPU memory with a context length of around $200\,000$, while NAtS could efficiently reduce the pre-filling and decoding latency and memory usage, allowing NAtS to do inference to a context length up to $700\,000$, 3.5 times larger than the full attention transformer. For the context length of $200\,000$, NAtS consumes $2.24\times$ less memory with a $3.0\times$ latency speed up during pre-filling and $2.6\times$ less memory with a $1.38\times$ decoding speed up.

## 5 Conclusion and Future Work

Efficiently managing the KV cache is crucial for deploying large language models in long-context applications. In this work, we propose NAtS, an approach that automatically optimizes the optimal roles of each token to determine how long they should survive. By constructing a learnable mask, NAtS learns to sparsify the attention map end-to-end. Our experiments show that NAtS uses much less KV caches compared to the State-of-the-art KV caching reduction approach. While NAtS shows promising results, future work could include exploration of further token roles and structured search spaces with hierarchies. Overall, we believe that NAtS paves the way towards more efficient and scalable inference with LLMs.

## 6 Acknowledgement

The authors gratefully acknowledge the computing time provided to them on the high-performance computers Noctua2 at the NHR Center PC2 under the project hpc-prf-intexml. These are funded by the Federal Ministry of Education and Research and the state governments participating on the basis of the resolutions of the GWK for the national high performance computing at universities (www.nhr-verein.de/unsere-partner).

Difan Deng was supported by the Federal Ministry of Education and Research (BMBF) under the project AI service center KISSKI (grantno.01IS22093C).

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

# A   Limitations

While NAtS shows promising results that efficiently reduce the KV cache sizes and our search space already contains most of the KV cache eviction strategies. However, the construction of the attention mask is still column-wise oriented and mostly focuses on the vertical style sparse attention [43]. Hence, this work does not involve the block-sparse attention and slash attention that are widely observed in other frameworks [43, 50]. Which in turn might result in a more sophisticated searching and inference process, as we can no longer easily drop the desired tokens since they can also be related to the value of the Q matrix. One potential future direction would be to introduce a more structured search space that allows for the construction of more sparse attention types.

# B   Broader Impact

LLMs are widely applied to different fields nowadays. However, the cost for LLM to store the KV cache and predict the next token is still huge, given the $\mathcal{O}(L)$ computation and memory costs of full Attention Models. This prevents further adaptation of LLMs (and other transformer-based foundation models with properties similar to those described before) because of high energy consumption and limited context windows. NAtS achieves a substantial reduction in KV cache size with minimal impact on model performance, outperforming existing state-of-the-art approaches. This increased efficiency can enable the deployment of larger, more powerful language models on resource-constrained devices and facilitate the development of new applications that rely on long-context understanding, such as advanced conversational AI, comprehensive document summarization, and complex code generation. By making long-context processing more accessible, NAtS has the potential to accelerate progress in natural language processing and related fields. Nevertheless, it does not solve other inherent problems of LLMs such as hallucinations.

# C   Details on Backward Propagation

## C.1   Gradients for Attention Masks

To compute the gradients for M, we set $g(A, M) = e^A \odot M$; then the gradient for M is:

$$\frac{\partial O}{\partial M} = \frac{\partial O}{\partial g} \frac{\partial g}{\partial M} \tag{8}$$

$$\frac{\partial g}{\partial M} = e^A \tag{9}$$

$$\frac{\partial g}{\partial A} = e^A \odot M \tag{10}$$

In Eq. 9 and 10, we show the gradient for $M$ is the same as the value that $\partial O/\partial A$ is supposed to be if no mask is applied. Since we have $g_{i,j} = e^{A_{i,j}} \odot M_{i,j}$. Let's set $S_i := \sum_j g_{i,j}$ and $P_{i,j} := \frac{g_{i,j}}{S_i}$. Then we have $\mathbf{dP} = \mathbf{dOV}^T$. Therefore,

$$dg_{i:} = (diag(\frac{1}{S_{i:}}) - \frac{1}{S_{i:}} P_{i:}^T) dP_{i:} \tag{11}$$

Combining Eq. 11 with Eq. 10 and Eq. 9 provides the same gradients as the vanilla softmax function with additive masks:

$$dM_{i:} = (diag(\frac{e^{A_{i:}}}{S_{i:}}) - \frac{e^{A_{i:}}}{S_{i:}} P_{i:}^T) dP_{i:} \tag{12}$$

$$dA = dM \odot M$$

$$= (diag(P_{i:}) - P_{i:} P_{i:}^T) dP_{i:} \tag{13}$$

Since $dg_{i:}$ is required to compute the gradient for $\mathbf{A}_{i:}$ and always needs to be computed. We can directly use this information to compute the gradients for the attention Mask $M$.

Following FlashAttention [19], we define $D_i = do_i^T o_i$, then

$$dM_{i,j} = \frac{e^{A_{i,j}}}{S_{i:}}(dP_{ij} - D_i) \tag{14}$$

$$dA_{i,j} = P_{i,j}(dP_{ij} - D_i) \tag{15}$$

and $dA$ is computed by Eq. 13. After that, we can backpropagate $dA$ to $dq$ and $dv$. Since $M$ needs to be recomputed anyway in the flash attention's backward process, this only results in little computational overhead.

However, in practice, we cannot guarantee an upper bound for $\frac{e^{A_{i,j}}}{S_i}$ with $M_{i,j} = 0$ since $S_i := \sum_j e^{A_{i,j}} \odot M_{i,j}$. As a result, $\lim_{e^{(A_{i,j})} \to \infty}(\frac{e^{(A_{i,j})}}{S_i}) = \infty$ when $M_{i,j} = 0$. Hence, we first clip $\frac{e^{A_{i,j}}}{S_i}$ within $(0, 1)$ and then compute $dM$ with the clipped value:

$$P'_{i,j} = \min(\frac{e^{A_{i,j}}}{S_i}, 1) \tag{16}$$

$$dM_{i,j} = P'_{i,j}(dP_{ij} - D_i) \tag{17}$$

### C.2   Details on computing the gradients for computing the token gradients info $d\alpha$

The gradients towards each token are collected through the column sum of each value weighted by the corresponding attention masks:

$$d\alpha_i = \sum_j dM_{i,j} \times M_{i,j}^\alpha \tag{18}$$

Where $\alpha \in \{G, L, SW\}$ is the discretized token type. Intuitively, this shows the model's preference over short-range or long-range correlations: If $i$ is quite close to $j$, then all the $\alpha$ will receive the same gradient information. However, if the model wants to create a long-range correlation with $i \gg j$, only Global Tokens will receive the gradient information. The network will therefore prefer to classify the corresponding tokens as global tokens.

Eq. 5 shows that the Global Token controls the local mask size. Therefore, the gradients for Global Token $i$ should also be regularized by the gradient information from the previous tokens:

$$\frac{\partial M_{i,j}^L}{\partial G_k} = - \prod_{\substack{n=j+1 \\ n \neq k}}^{i-1} (1 - G_n) \tag{19}$$

In cases where $G_k$ is 0, Equation 19 is the negative value of Equation 5. However, for the case where $G_k = 1$, this is equivalent to the local mask values where $k$ is no longer set as a Global Token. This requires us to find the index of the next global token $I_{k+1}^{G_{\text{next}}}$ and the last global token $I_{k-l}^{G_{\text{next}}}$ where $l$ is the length of the local sequence that ends at $k$.

This gradient information will then be collected and subtracted from the computed Global Tokens gradient values:

$$d\alpha_i^{G-} = \begin{cases} \sum_{\substack{m \geq i \\ i > n \geq I_{k-l}^{G_{\text{next}}}}} M_{m,n}^L \times dM_{m,n} & \text{if } G_i = 0, \\ \sum_{\substack{I_{i+1}^{G_{\text{next}}} \geq m >= i \\ i > n \geq I_{k-l}^{G_{\text{next}}}}} dM_{m,n} & \text{if } G_i = 1, \end{cases} \tag{20}$$

$$d\alpha_i^G = d\alpha_i^G - d\alpha_i^{G-} \tag{21}$$

Intuitively, this gradient term $d\alpha_i^{G-}$ checks if the new Global Token needs to be inserted into the current sub-sequence (when $G$ is 0) or we should remove the current Global Token to enlarge the current sub-sequence (when $G$ is 1). We further illustrate this process in the appendix.

Figure 5 illustrates an example of this. Token 4 is a global token, and we search for its next Global Token (which is Token 10 in this example). Assuming that we want to change Token 4 to another

role, the regions within the boundary (orange ones) are those tokens that are influenced by this swtching: given that Token 4 no longer becomes Global Token, Token 2 and 3 will be part of a larger subsequence that ranges from 1 to 10, and the attention maps within the orange region should not be masked out. Intuitively, this gradient measures the regret that we made in order to switch one Global Token into a Local Token. A similar idea can be found in the red region. Assuming that we want to switch Token 7 to a Global Token, then Tokens 5,6 will be split into a new subsequence and we will no longer connect them with Tokens 8, 9, 10 since they belong to two different sub-sequences after the switch. Hence, this value in the red regions measures the regret if we mistakenly classify a Global Token as non-Global Tokens.

In practice, the $d\alpha_i^{G-}$ values for Global Token can be easily collected by looking at the last and next $G$ index with a scan function to collect the gradients to the corresponding positions. However, computing $d\alpha_i^{G-}$ with $G_i = 0$ requires the gradient information from all the past token mask information. This might be computationally prohibitive in practice, since the valid values contained in Equation 20 are sparse, we ignore the cases for $G_i = 0$ and only compute the dense gradients for the case where $G_i = 1$.

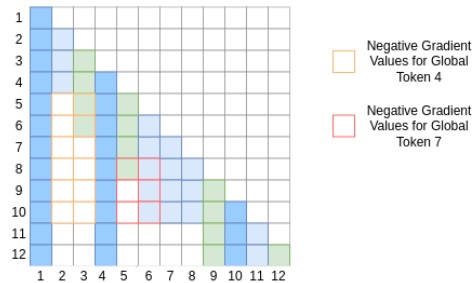

Figure 5: The gradient term $d\alpha_i^{G-}$ for Token 4 and 7

### C.3 Optimizing for sparser Attention Maps

The sparse regularization term $\lambda$ is directly applied to the corresponding gradients for the Global Tokens and Local Tokens. Intuitively, if the column-wise sum of the gradients for each iteration is smaller than $\lambda$, the attention maps of that column would require no further update or gradients updates towards a sparser transformer, they will be push the projection layer to classify those tokens as sliding window tokens. This ensures that the global tokens and local tokens actually require a certain amount of column-wise attention map values to keep them activated. Hence, the sparse regularization term $\lambda$ could also be considered as a soft threshold for the attention map values, where attention map values smaller than that will be encouraged to be filtered out. This value should therefore penalize the number of unmasked tokens for each row. While the number of unmasked tokens for Global Token and Local Tokens in column $i$ are $L - i$ and $I_i^{G_{\text{next}}} - i$ with $I_i^{G_{\text{next}}}$ defined in Section 3.2. Hence, we have:

$$dG_i^{G_{sparse}} = \lambda \times \frac{L - i}{L} \tag{22}$$

$$dG_i^{L_{sparse}} = \lambda \times \frac{I_i^{G_{\text{next}}} - i}{L} \tag{23}$$

Combining Eq. 18, 20, 22 and Eq. 23, we have:

$$d\alpha_i^G = \sum_j dM_{i,j} \times M_{i,j}^G \times M_{i,j}^{casual} + dG_i^{G_{sparse}} - d\alpha_i^{G-} \tag{24}$$

$$d\alpha_i^L = \sum_j dM_{i,j} \times M_{i,j}^L \times M_{i,j}^{casual} + dG_i^{L_{sparse}} \tag{25}$$

$$d\alpha_i^{SW} = \sum_j dM_{i,j} \times M_{i,j}^{SW} \times M_{i,j}^{casual} \tag{26}$$

with $M_{i,j}^G, M_{i,j}^{SW}, M_{i,j}^L$ defined in Eq. 3, 6 and 4 and $M_{i,j}^{casual}$ is a casual attention mask.

## D  Integrating NAtS into FlashAttention

Flashattention [18] continuously loads Q, K, V values to the SRAM and only computes the attention maps within the loaded blocks. NAtS loads the corresponding token states information $\alpha$ and construct the masks $M$ dynamically with Equation 3, 5, 4. During the backward process, Flashattention collects the row-wise gradients for computing $d\mathbf{Q}$ and column-wise gradients to compute $d\mathbf{K}$ and $d\mathbf{V}$. Hence, the gradients $d\alpha$ can be computed together with $d\mathbf{K}$ and $d\mathbf{V}$. We show how NAtS can be

---

**Algorithm 1** NAtS forward pass on FlashAttention2, we mark the NAtS related operations with blue

---

**Require:** Matrices $\mathbf{Q}, \mathbf{K}, \mathbf{V} \in \mathbb{R}^{L \times d}$, token states $\alpha \in \mathbb{R}^{L \times N_{\text{opts}}}$, indices of the next token states $\mathbf{I}^{\mathbf{G}_{\text{next}}} \in \mathbb{R}^{N}$ in HBM, block sizes $B_c, B_r$, sliding window size $W$

1: Divide $\mathbf{Q}$ into $T_r = \left\lceil \frac{N}{B_r} \right\rceil$ blocks $\mathbf{Q}_1, \ldots, \mathbf{Q}_{T_r}$ of size $B_r \times d$ each, and divide $\mathbf{K}, \mathbf{V}, \alpha, \mathbf{I}^{\mathbf{G}_{\text{next}}}$ into $T_c = \left\lceil \frac{N}{B_c} \right\rceil$ blocks $\mathbf{K}_1, \ldots, \mathbf{K}_{T_c}, \mathbf{V}_1, \ldots, \mathbf{V}_{T_c}$, of size $B_c \times d$, $\alpha_1, \ldots, \alpha_{T_c}$ of size $B_r \times N_{\text{opts}}$, and $\mathbf{I}^{\mathbf{G}_{\text{next}}}{}_1, \ldots, \mathbf{I}^{\mathbf{G}_{\text{next}}}_{\mathbf{T_c}}$ of size $B_r$ each.

2: Divide the output $\mathbf{O} \in \mathbb{R}^{N \times d}$ into $T_r$ blocks $\mathbf{O}_i, \ldots, \mathbf{O}_{T_r}$ of size $B_r \times d$ each, and divide the logsumexp $L$ into $T_r$ blocks $L_i, \ldots, L_{T_r}$ of size $B_r$ each.

3: **for** $1 \leq i \leq T_r$ **do**
4:     Load $\mathbf{Q}_i$ from HBM to on-chip SRAM.
5:     On chip, initialize $\mathbf{O}_i^{(0)} = (0)_{B_r \times d} \in \mathbb{R}^{B_r \times d}, \ell_i^{(0)} = (0)_{B_r} \in \mathbb{R}^{B_r}, m_i^{(0)} = (-\infty)_{B_r} \in \mathbb{R}^{B_r}$.

6:     **for** $1 \leq j \leq T_c$ **do**
7:         Load $\alpha_j$ and $\mathbf{I}^{\mathbf{G}_{\text{next}}}{}_j$ to SRAM and check if the current block contain any valid values according to Equations 3,4,6,
8:         **if** Do Compute **then**
9:             Construct attention mask $\mathbf{M}_{i,j}$ with $\alpha_j$, $\mathbf{I}^{\mathbf{G}_{\text{next}}}{}_j$, and $W$ with Equations 3,4,6
10:            Load $\mathbf{K}_j, \mathbf{V}_j$ from HBM to on-chip SRAM.
11:            On chip, compute $\mathbf{S}_i^{(j)} = \mathbf{Q}_i \mathbf{K}_j^T \in \mathbb{R}^{B_r \times B_c}$.
12:            On chip, compute $m_i^{(j)} = \max(m_i^{(j-1)}, \text{rowmax}(\mathbf{S}_i^{(j)})) \in \mathbb{R}^{B_r}, \tilde{\mathbf{P}}_i^{(j)} = \exp(\mathbf{S}_i^{(j)} - m_i^{(j)}) \textcolor{blue}{\odot \mathbf{M}_{i,j}} \in \mathbb{R}^{B_r \times B_c}$ (pointwise), $\ell_i^{(j)} = e^{m_i^{j-1} - m_i^{(j)}} \ell_i^{(j-1)} + \text{rowsum}(\tilde{\mathbf{P}}_i^{(j)}) \in \mathbb{R}^{B_r}$.

13:            On chip, compute $\mathbf{O}_i^{(j)} = \text{diag}(e^{m_i^{(j-1)} - m_i^{(j)}})^{-1} \mathbf{O}_i^{(j-1)} + \tilde{\mathbf{P}}_i^{(j)} \mathbf{V}_j$.
14:        **end if**
15:    **end for**
16:    On chip, compute $\mathbf{O}_i = \text{diag}(\ell_i^{(T_c)})^{-1} \mathbf{O}_i^{(T_c)}$.
17:    On chip, compute $L_i = m_i^{(T_c)} + \log(\ell_i^{(T_c)})$.
18:    Write $\mathbf{O}_i$ to HBM as the $i$-th block of $\mathbf{O}$.
19:    Write $L_i$ to HBM as the $i$-th block of $L$.
20: **end for**
21: Return the output $\mathbf{O}$ and the logsumexp $L$.

---

integrated into Flashattention2 [18] in Algorithms 1 and 2. We highlight the difference between the vanilla Flashattention2 and Flashattention2 with NAtS in blue. Compared with vanilla Flashattention2 models, NAtS adaptively skips the blocks that do not have valid mask values to avoid unnecessary computation. These invalid blocks will only be applied to update the gradients for $d\alpha$.

## E  Detailed Inference Process

In Section 3.3, we showed that NAtS could efficiently update the KV cache values and drop the unnecessary tokens. Here we provide a detailed pseudocode for this updating process.

As shown in Algorithm 3, once we receive a new KV cache pair, we first check its corresponding type. Depending on the new token type, we either: 1. append the new KV values after the next global tokens and remove the remaining local tokens 2. append the new KV values after the existing tokens 3. put the new KV values to the tail of the sliding window token queues (located at the beginning of the KV cache values)

## F  Experiments Details

Here we discuss further details in our experiments. The codes for NAtS can be found under `https://github.com/automl/NeuralAttentionSearch`

**Algorithm 2** NAtS backward pass on FlashAttention2, we mark the NAtS related operations with blue

**Require:** Matrices $\mathbf{Q}, \mathbf{K}, \mathbf{V}, \mathbf{O}, \mathbf{dO} \in \mathbb{R}^{L \times d}$, token states $\alpha \in \mathbb{R}^{L \times N_{\text{opts}}}$, indices of the next token states $\mathbf{I}^{\mathbf{G}_{\text{next}}} \in \mathbb{R}^{L}$ in HBM, vector $L \in \mathbb{R}^{N}$ in HBM, block sizes $B_c$, $B_r$, sliding window size $W$.

1: Divide $\mathbf{Q}$ into $T_r = \left\lceil \frac{N}{B_r} \right\rceil$ blocks $\mathbf{Q}_1, \dots, \mathbf{Q}_{T_r}$ of size $B_r \times d$ each, and divide $\mathbf{K}, \mathbf{V}, \alpha, \mathbf{I}^{\mathbf{G}_{\text{next}}}$ in to $T_c = \left\lceil \frac{N}{B_c} \right\rceil$ blocks $\mathbf{K}_1, \dots, \mathbf{K}_{T_c}$ and $\mathbf{V}_1, \dots, \mathbf{V}_{T_c}$, of size $B_c \times d$, $\alpha_1, \dots, \alpha_{T_c}$ of size $B_r \times N_{\text{opts}}$, and $\mathbf{I}^{\mathbf{G}_{\text{next}}}{}_1, \dots, \mathbf{I}^{\mathbf{G}_{\text{next}}}_{\mathbf{T_c}}$ of size $B_r$ each.

2: Divide $\mathbf{O}$ into $T_r$ blocks $\mathbf{O}_i, \dots, \mathbf{O}_{T_r}$ of size $B_r \times d$ each, divide $\mathbf{dO}$ into $T_r$ blocks $\mathbf{dO}_i, \dots, \mathbf{dO}_{T_r}$ of size $B_r \times d$ each, and divide $L$ into $T_r$ blocks $L_i, \dots, L_{T_r}$ of size $B_r$ each.

3: Initialize $\mathbf{dQ} = (0)_{N \times d}$ in HBM and divide it into $T_r$ blocks $\mathbf{dQ}_1, \dots, \mathbf{dQ}_{T_r}$ of size $B_r \times d$ each. Divide $\mathbf{dK}, \mathbf{dV} \in \mathbb{R}^{N \times d}$ in to $T_c$ blocks $\mathbf{dK}_1, \dots, \mathbf{dK}_{T_c}$ and $\mathbf{dV}_1, \dots, \mathbf{dV}_{T_c}$, of size $B_c \times d$ each. Divide $d\alpha \in \mathbb{R}^{N_{\text{opts}}}$ in to $d\alpha_1, \dots, d\alpha_{T_c}$ of size $B_r \times N_{\text{opts}}$ each.

4: Compute $D = \text{rowsum}(\mathbf{dO} \odot \mathbf{O}) \in \mathbb{R}^d$ (pointwise multiply), write $D$ to HBM and divide it into $T_r$ blocks $D_1, \dots, D_{T_r}$ of size $B_r$ each.

5: **for** $1 \leq j \leq T_c$ **do**

6:     Load $\mathbf{K}_j, \mathbf{V}_j, \alpha_j$ and $\mathbf{I}^{\mathbf{G}_{\text{next}}}{}_j$ from HBM to on-chip SRAM.

7:     Initialize $\mathbf{dK}_j = (0)_{B_c \times d}, \mathbf{dV}_j = (0)_{B_c \times d}, d\alpha_j = (0)_{B_c \times N_{\text{opts}}}$ on SRAM.

8:     **for** $1 \leq i \leq T_r$ **do**

9:         Load $\mathbf{Q}_i, \mathbf{O}_i, \mathbf{dO}_i, \mathbf{dQ}_i, L_i, D_i$ from HBM to on-chip SRAM.

10:         On chip, compute $\mathbf{S}_i^{(j)} = \mathbf{Q}_i \mathbf{K}_j^T \in \mathbb{R}^{B_r \times B_c}$.

11:         On chip, Construct attention mask $\mathbf{M}_{i,j}$ with $\alpha_j$, $\mathbf{I}^{\mathbf{G}_{\text{next}}}{}_j$, and $W$ with Equations 3,4,6

12:         On chip, compute $\mathbf{P}'^{(j)}_i = \exp(\mathbf{S}_{ij} - L_i) \in \mathbb{R}^{B_r \times B_c}$ and $\mathbf{P}^{(j)}_i = \mathbf{P}'^{(j)}_i \odot \mathbf{M}_{i,j}$.

13:         Check if the current block contains any valid value

14:         **if** Do compute **then**

15:           On chip, compute $\mathbf{dV}_j \leftarrow \mathbf{dV}_j + (\mathbf{P}^{(j)}_i)^\top \mathbf{dO}_i \in \mathbb{R}^{B_c \times d}$.

16:           On chip, compute $\mathbf{dP}^{(j)}_i = \mathbf{dO}_i \mathbf{V}_j^\top \in \mathbb{R}^{B_r \times B_c}$.

17:           On chip, compute $\mathbf{dS}^{(j)}_i = \mathbf{P}^{(j)}_i \odot (\mathbf{dP}^{(j)}_i - D_i) \in \mathbb{R}^{B_r \times B_c}$.

18:           Load $\mathbf{dQ}_i$ from HBM to SRAM, then on chip, update $\mathbf{dQ}_i \leftarrow \mathbf{dQ}_i + \mathbf{dS}^{(j)}_i \mathbf{K}_j \in \mathbb{R}^{B_r \times d}$, and write back to HBM.

19:           On chip, compute $\mathbf{dK}_j \leftarrow \mathbf{dK}_j + \mathbf{dS}^{(j)}_i{}^\top \mathbf{Q}_i \in \mathbb{R}^{B_c \times d}$.

20:         **end if**

21:         On chip, clip $\mathbf{P}'^{(j)}_i$ to $(0,1)$ and compute $d\mathbf{S}'^{(j)}_i = \mathbf{P}'^{(j)}_i \odot (\mathbf{dP}^{(j)}_i - D_i) \in \mathbb{R}^{B_r \times B_c}$

22:         On chip, update $d\alpha_j$ with Equations 24, 25, 26

23:     **end for**

24:     Write $\mathbf{dK}_j, \mathbf{dV}_j$, and $d\alpha_j$ to HBM.

25: **end for**

26: Return $\mathbf{dQ}, \mathbf{dK}, \mathbf{dV}$.

## F.1 Detailed Hyperparameter Setting for GPT2-small Training

Following the setting from NanoGPT [45], the GPT-2 style small has 12 layers and 12 heads with a hidden dimension of 768. Instead of the learnable position encoding, we apply rotary embeddings [74] to each transformer layer. The PG-19 dataset contains books extracted from Project Gutenberg [67] with about 2B tokens in the training sets. We train all models with a context length of 1024 and a batch size of 480 (using gradient accumulation). We train them for 600 000 iterations evaluate them on the test sets of PG19 with a context length of 1024. Training one model can be finished in 16 hours with 4 Nvidia H100 PCEi GPUs.

## F.2 Collecting the Fine-tune Training Set

To fine-tune NAtS on LLMs, we collect the training datasets from different tasks for real-world tasks:

**Algorithm 3** NAtS KV Cache Updating Process

---

**Require:** KV values $\mathbf{K}, \mathbf{V} \in \mathbb{R}^{N \times H \times d_{head}}$, sliding window size $W$, $\alpha \in \mathbb{R}^{L \times N_{\text{opts}}}$. Existing KV cache $\mathbf{K}^{Cache}, \mathbf{V}^{Cache} \in \mathbb{R}^{W \times H \times d_{heads}}$, number of global tokens $n^{global} \in \mathbb{R}^{H}$, number of local tokens $n^{local} \in \mathbb{R}^{H}$, current tail of the sliding window queue $t$

1:  **for** $0 \leq i \leq L$ **do**
2:     **for** $0 \leq h \leq H$ **do**
3:        remove the tail of the sliding window queue: $\mathbf{K}_t^{Cache} = 0, \mathbf{V}_t^{Cache} = 0$
4:        **if** $\alpha_i$ is $GlobalToken$ **then**
5:           move $\mathbf{K}_{i,h}, \mathbf{V}_{i,h}$ to the $(n_h^{global})$th position of the KV cache values: $\mathbf{K}_{n_h^{global},h}^{Cache} = \mathbf{K}_{i,h}, \mathbf{V}_{n_h^{global},h}^{Cache} = \mathbf{V}_{i,h},.$
6:           update number of global and local tokens $n_h^{global} = n_h^{global} + 1, n_h^{local} = 0$
7:           remove all the KV cache values after $n_h^{global}$: $\mathbf{K}_{j,h}^{Cache} = 0, \mathbf{V}_{j,h}^{Cache} = 0, for\ j > n_h^{global}$
8:        **else if** $\alpha_i$ is $LocalToken$ **then**
9:           move $\mathbf{K}_{i,h}, \mathbf{V}_{i,h}$ to the $(n_h^{global} + n_h^{local})$th position of the KV cache values: $\mathbf{K}_{n_h^{global}+n_h^{local},h}^{Cache} = \mathbf{K}_{i,h}, \mathbf{V}_{n_h^{global}+n_h^{local},h}^{Cache} = \mathbf{V}_{i,h},.$
10:         update number of local tokens $n_h^{local} = n_h^{local} + 1$
11:        **else**
12:          move $\mathbf{K}_{i,h}, \mathbf{V}_{i,h}$ to the $(t)$th position of the KV cache values: $\mathbf{K}_{t,h}^{Cache} = \mathbf{K}_{i,h}, \mathbf{V}_{t_h+n_h^{local},h}^{Cache} = \mathbf{V}_{i,h},.$
13:        **end if**
14:        generate masks for the corresponding valid values
15:        move the queue tail to the next value: $t = (t + 1) \bmod W$
16:     **end for**
17: **end for**

---

- Multi-Document QA: HotPotQA [88], 2WikiMultihopQA [37], MuSiQue [78], and DuReader (zh) [36]
- Single-Document QA: NarrativeQA [48] and Qasper [20]
- Summarization: GovReport [39], QMSum [97], MultiNews [25], and VCSUM (zh) [82]
- Few-shot Leraning: TREC [53], TriviaQA [44], and SAMSum [33]
- Code Completion: LCC [35] and RepoBench-P [56]

We also construct the synthetic passkey-retrieval dataset introduced in DuoAttention [85]. This dataset is generated by embedding multiple random passkeys in different locations with a long context. The model will then be asked to recall this passkey information.

For all the few-shot learning datasets, following Bai et al. [12], we randomly concatenate multiple question-answer pairs into one single extended context as one piece of data. The number of con-catenated samples for the TREC dataset ranges from $[10, 100]$. This value is $[2, 6]$ for TriviaQA and $[10, 50]$ for SAMSum. Additionally, for the datasets that do not have enough context length (e.g., the DuReader dataset), we also merge multiple documents into one piece of data (in our case, this value is 4).

We do not collect all the data whose length goes beyond a threshold to ensure that the context can be fitted into our model. Additionally, we collect at most 500 instances in each dataset since some datasets might not contain data. In the end, the real-world dataset contains 6436 data instances. We add another 564 instances from the synthetic dataset. In the end, our dataset contains 7000 instances. Finetuning a model on this dataset for one epoch takes roughly 8 hours with 2 Nvidia H100 PCIe GPUs.

|  | Full | Duo | SLLM | H2O | Snap | Ada | Chunk | Critical | Pyradmid | MoA | NAtS |
|---|---|---|---|---|---|---|---|---|---|---|---|
| 4k | 93.68 | 92.63 | 58.89 | 5.60 | 46.50 | 57.92 | 59.33 | 74.06 | 44.00 | 55.73 | **93.68** (49%) |
| 8k | 91.29 | 89.94 | 59.21 | 2.82 | 38.12 | 49.30 | 53.53 | 83.12 | 37.66 | 41.98 | **90.76** (43%) |
| 16k | 89.85 | 87.51 | 55.18 | 3.00 | 38.59 | 52.80 | 63.46 | 82.83 | 38.05 | 34.31 | **89.09** (41%) |
| 32k | 81.24 | 78.81 | 47.63 | 2.70 | 49.32 | 66.20 | 71.91 | 76.23 | 46.16 | 27.92 | **80.39** (42%) |

|  | Full | Duo | SLLM | H2O | Snap | Ada | Chunk | Critical | Pyradmid | MoA | NAtS |
|---|---|---|---|---|---|---|---|---|---|---|---|
| 4k | 93.68 | 52.44 | 42.96 | 3.26 | 33.72 | 38.79 | 42.74 | 46.11 | 33.66 | 33.26 | **93.18** (29%) |
| 8k | 91.29 | 40.96 | 36.54 | 2.37 | 31.36 | 36.67 | 42.01 | 46.73 | 28.64 | 25.47 | **89.97** (24%) |
| 16k | 89.85 | 35.42 | 33.38 | 1.92 | 30.30 | 37.48 | 42.84 | 57.17 | 28.17 | 25.55 | **87.69** (21%) |
| 32k | 81.24 | 40.00 | 32.76 | 1.70 | 35.17 | 44.46 | 60.56 | 61.55 | 31.84 | 24.88 | **78.60** (21%) |

Table 3: Ruler results with 50% KV budgets (top) and 25% KV budget (bottom) on Mistral-7B-Instruct-v0.3. All Full models use the 100% KV budgets. We mark the actual KV budgets used by NAtS in the bracket.

# G    Further Experimental Results

## G.1    Results on Ruler Benchmark with Mistral

Table 3 shows the results of different optimizers on the Mistral-7B-Instruct-v0.3 model. In this case, the $\lambda$ for 25% and 50% budgets are $1e-6$ and $3e-7$ respectively. We found that to achieve the same sparsity level, Mistral-7B requires a larger sparse regularization value $\lambda$ compared to the Llama 3.1 model. This might indicate that the attention map values in Mistral are more evenly distributed, and thus we need a larger $\lambda$ to force more tokens to only focus on the local information. The maximal context length of Mistral 7B is $32k$. We only evaluate the models with a maximal length of $32k$. The result is consistent with the results shown in Table 1. NAtS consistently outperforms the other baselines with smaller budgets.

## G.2    Further results on LongBench

We show additional results on fine-tuning LLM on the LongBench dataset here. Table 4 shows the results with 25% KV budgets on Mistral-7B-Instruct-v0.3 with $\lambda = 3e-6$. The result confirms our conclusion that NAtS outperforms the other baselines in most datasets under a similar budget level (and many times with even smaller budgets).

|  | Full | Duo | SLLM | H2O | Snap | Ada | Chunk | Critical | Pyradmid | MoA | NAtS |
|---|---|---|---|---|---|---|---|---|---|---|---|
| NarrativeQA | 29.21 | 16.49 | 27.56 | 23.65 | 22.70 | 24.34 | 21.22 | **27.72** | 21.97 | 21.67 | 26.7(9%) |
| Qasper | 41.34 | 14.58 | 29.16 | 18.95 | 23.08 | 22.99 | 21.21 | 28.85 | 23.54 | 19.72 | **41.46**(21%) |
| MultiFieldQA-en | 52.51 | 29.83 | 31.88 | 31.52 | 38.56 | 38.11 | 34.11 | 45.18 | 35.34 | 28.80 | **52.36**(18%) |
| MultiFieldQA-zh | 58.03 | 29.15 | 30.47 | 29.21 | 31.42 | 32.30 | 32.40 | 38.12 | 29.67 | 23.20 | **54.13**(16%) |
| HotpotQA | 49.62 | 34.85 | 46.07 | 42.10 | 46.37 | 48.83 | 46.93 | 47.48 | 45.94 | 39.64 | **51.45**(15%) |
| 2WikiQA | 40.01 | 27.83 | 35.11 | 30.43 | 33.22 | 33.14 | 32.03 | **38.24** | 33.96 | 27.02 | 37.05(18%) |
| Musique | 28.44 | 13.03 | 22.65 | 16.99 | 22.45 | 21.28 | 22.62 | 25.01 | 23.67 | 16.22 | **27.88**(14%) |
| DuReader (zh) | 34.87 | 26.51 | 18.69 | 20.66 | 25.34 | 27.55 | 28.45 | 27.04 | 24.82 | 19.11 | **35.25**(12%) |
| GovReport | 34.94 | 22.09 | 27.44 | 27.11 | 28.31 | 27.94 | 28.94 | 29.33 | 26.94 | 25.58 | **33.39**(14%) |
| QMSum | 25.67 | 17.25 | 22.11 | 20.84 | 21.43 | 21.95 | 21.95 | 22.77 | 22.39 | 19.53 | **24.70**(13%) |
| MultiNews | 27.85 | 23.41 | 22.01 | 23.03 | 23.77 | 23.62 | 23.34 | 24.37 | 23.89 | 22.55 | **27.35**(28%) |
| VCSUM (zh) | 16.34 | 13.58 | 15.41 | 15.13 | 15.28 | 15.38 | 14.69 | **16.10** | 14.96 | 14.87 | 16.06(11%) |
| TREC | 75.50 | 52.00 | 70.50 | 61.50 | 58.50 | 59.00 | 57.00 | 67.50 | 58.00 | 63.00 | **73.50**(20%) |
| TriviaQA | 88.89 | 84.85 | 88.92 | 86.60 | 88.89 | 89.39 | 89.19 | 88.98 | 89.31 | 86.94 | **89.80**(17%) |
| SAMSum | 47.29 | 42.15 | 45.44 | 45.16 | 46.40 | 46.72 | 46.51 | 46.67 | 46.58 | 43.90 | **46.91**(13%) |
| LSHT | 39.75 | 17.50 | 29.00 | 22.00 | 38.50 | 39.00 | **39.50** | 39.25 | 37.50 | 17.25 | 38.5(11%) |
| Passage Count | 5.50 | 3.50 | 6.00 | 6.00 | 6.00 | **8.00** | 5.50 | 6.00 | 6.50 | 5.00 | 4.5(14%) |
| PassageRetrieval-en | 98.00 | 62.50 | 86.50 | 63.00 | 91.50 | 94.00 | 87.50 | 89.50 | 88.00 | 17.25 | **96.00**(13%) |
| PassageRetrieval-zh | 96.50 | 10.50 | 26.50 | 34.50 | 69.25 | 83.25 | 71.00 | 90.00 | 75.50 | 16.00 | **92.00**(14%) |
| LCC | 53.02 | 50.87 | 51.22 | 52.77 | 55.67 | 55.67 | 53.76 | **56.17** | 55.46 | 51.08 | 53.81(29%) |
| RepoBench-P | 56.83 | 48.93 | 54.80 | 55.74 | 56.82 | **57.06** | 56.75 | 56.75 | 56.40 | 53.59 | 56.7(23%) |

Table 4: LongBench Results with 25% Budget Size on Mistral-7B. .

Tables 5 and 6 show the evaluation results on LongBench with 50% budgets. Despite having fewer KV cache budgets in all the datasets, NAtS still achieves comparable performance on the LLama3-8B model and better results on the Mistral model and generally performs comparable to the results with the full attention transformers.

| | Full | Duo | SLLM | H2O | Snap | Ada | Chunk | Critical | Pyradmid | MoA | NAtS |
|---|---|---|---|---|---|---|---|---|---|---|---|
| NarrativeQA | 31.35 | 29.40 | 28.18 | 20.15 | 30.37 | 30.04 | 29.49 | **32.03** | 29.18 | 13.92 | 31.39(29%) |
| Qasper | 24.73 | 20.37 | 18.15 | 18.09 | 18.44 | 19.42 | 17.24 | 22.13 | 18.54 | 22.25 | **25.17**(45%) |
| MultiFieldQA-en | 29.46 | 26.70 | 17.89 | 22.50 | 23.21 | 24.62 | 22.55 | 26.30 | 22.95 | 14.24 | **28.94**(41%) |
| MultiFieldQA-zh | 60.01 | 60.31 | 42.19 | 38.88 | 47.66 | 50.23 | 44.54 | 56.22 | 46.50 | 23.91 | **61.27**(37%) |
| HotpotQA | 17.06 | **18.84** | 16.73 | 15.61 | 16.55 | 16.29 | 17.22 | 15.83 | 15.86 | 8.85 | 17.28(37%) |
| 2WikiQA | 16.64 | 16.30 | 14.28 | 11.71 | 15.82 | 15.38 | 14.67 | **17.01** | 14.81 | 10.58 | 16.84(41%) |
| Musique | 11.59 | **13.77** | 10.75 | 10.12 | 10.78 | 11.19 | 10.63 | 12.85 | 10.64 | 5.03 | 11.89(36%) |
| DuReader (zh) | 35.56 | 33.29 | 18.16 | 27.32 | 29.22 | 30.82 | 28.49 | 32.81 | 27.93 | 22.80 | **33.36**(32%) |
| GovReport | 34.30 | 32.99 | 30.84 | 29.46 | 31.34 | 31.51 | 31.92 | 33.09 | 30.54 | 25.45 | **34.23**(36%) |
| QMSum | 23.30 | **23.89** | 21.43 | 20.72 | 22.08 | 22.74 | 22.22 | 22.89 | 22.92 | 19.76 | 22.6(33%) |
| MultiNews | 27.13 | 26.29 | 24.77 | 25.10 | 25.27 | 25.48 | 25.08 | 25.81 | 25.10 | 24.86 | **27.17**(51%) |
| VCSUM (zh) | 16.36 | 15.70 | 15.29 | 15.03 | 15.70 | 15.77 | 15.59 | 15.89 | 15.32 | 14.67 | **16.24**(28%) |
| TREC | 72.50 | 72.50 | 71.00 | 65.50 | 60.00 | 66.50 | 63.50 | 70.50 | 59.50 | 64.00 | **73.50**(44%) |
| TriviaQA | 91.15 | 90.41 | 91.44 | 90.25 | 91.47 | 90.97 | 91.63 | 91.14 | 90.98 | 81.17 | **92.14**(40%) |
| SAMSum | 43.72 | 42.68 | 43.41 | 42.57 | **44.21** | 43.44 | 43.00 | 43.41 | 43.91 | 39.64 | 42.61(30%) |
| LSHT | 46.50 | **46.50** | 40.00 | 34.50 | 45.00 | 46.00 | 45.00 | 46.50 | 45.00 | 24.00 | 45.5(31%) |
| Passage Count | 6.63 | 6.67 | 6.73 | 2.21 | 6.33 | 6.07 | 6.03 | 4.74 | 8.04 | 1.80 | **9.78**(37%) |
| PassageRetrieval-en | 97.98 | **98.55** | 96.67 | 93.54 | 95.98 | 97.88 | 95.83 | 98.12 | 96.10 | 17.96 | 97.32(36%) |
| PassageRetrieval-zh | 77.99 | 75.58 | 43.61 | 67.83 | **79.66** | 78.27 | 72.38 | 78.22 | 79.48 | 19.72 | 78.17(34%) |
| LCC | 54.10 | **56.22** | 53.10 | 55.61 | 54.08 | 53.13 | 54.23 | 52.33 | 51.87 | 52.28 | 53.27(57%) |
| RepoBench-P | 51.39 | **57.54** | 50.80 | 55.17 | 52.49 | 51.59 | 52.32 | 52.38 | 51.59 | 48.46 | 52.68(47%) |

Table 5: LongBench Results with 50% Budget Size on LLama 3.1 8B .

| | Full | Duo | SLLM | H2O | Snap | Ada | Chunk | Critical | Pyradmid | MoA | NAtS |
|---|---|---|---|---|---|---|---|---|---|---|---|
| NarrativeQA | 29.21 | 26.95 | **28.76** | 24.00 | 24.65 | 25.40 | 25.66 | 28.20 | 24.41 | 23.58 | 28.1(31%) |
| Qasper | 41.34 | 35.65 | 35.80 | 27.88 | 31.68 | 31.94 | 31.38 | 38.10 | 31.50 | 28.60 | **41.75**(46%) |
| MultiFieldQA-en | 52.51 | 52.50 | 37.35 | 40.15 | 46.12 | 48.52 | 44.73 | 52.20 | 45.70 | 35.52 | **52.79**(42%) |
| MultiFieldQA-zh | 58.03 | 55.37 | 36.43 | 39.43 | 40.55 | 40.92 | 43.70 | 47.93 | 38.24 | 27.48 | **56.75**(38%) |
| HotpotQA | 49.62 | **52.94** | 47.66 | 47.24 | 49.34 | 49.57 | 49.51 | 47.20 | 51.26 | 39.64 | 50.73(39%) |
| 2WikiQA | 40.01 | 39.26 | 38.44 | 38.09 | 37.27 | 36.75 | 40.24 | 38.13 | 37.49 | 31.42 | **40.77**(42%) |
| Musique | 28.44 | **29.45** | 27.24 | 20.61 | 26.43 | 25.77 | 27.28 | 28.54 | 25.85 | 16.84 | 28.64(38%) |
| DuReader (zh) | 34.87 | **36.44** | 18.63 | 26.82 | 30.29 | 32.18 | 32.72 | 32.18 | 29.71 | 20.79 | 34.7(33%) |
| GovReport | 34.94 | 32.07 | 30.79 | 30.58 | 31.43 | 31.55 | 32.16 | 32.39 | 29.63 | 25.80 | **34.23**(38%) |
| QMSum | 25.67 | 24.20 | 22.91 | 22.48 | 23.16 | 24.36 | 23.47 | 25.12 | 24.11 | 21.08 | **25.94**(37%) |
| MultiNews | 27.85 | 26.97 | 25.18 | 25.62 | 26.45 | 26.24 | 26.26 | 26.26 | 25.56 | 24.39 | **27.47**(54%) |
| VCSUM (zh) | 16.34 | 15.58 | 16.24 | 15.80 | 15.94 | 16.42 | 15.73 | **16.57** | 15.63 | 14.13 | 16.55(31%) |
| TREC | 75.50 | 74.50 | 74.00 | 66.00 | 66.50 | 68.50 | 69.00 | 74.50 | 64.50 | 69.00 | **75.00**(49%) |
| TriviaQA | 88.89 | 87.37 | 89.06 | 88.36 | 88.71 | 88.81 | 88.69 | 88.08 | **89.55** | 87.05 | 89.3(41%) |
| SAMSum | 47.29 | 44.85 | 47.30 | 45.82 | 47.19 | 47.30 | 46.77 | **47.31** | 46.85 | 45.29 | 46.87(35%) |
| LSHT | 39.75 | 37.50 | 33.00 | 31.50 | 39.25 | 39.25 | 39.25 | **40.25** | 38.00 | 21.00 | 38.75(32%) |
| Passage Count | 5.50 | 6.00 | 5.50 | 4.00 | 4.00 | 5.50 | 5.00 | 5.00 | 4.00 | 4.50 | **7.50**(39%) |
| PassageRetrieval-en | 98.00 | **99.00** | 89.50 | 81.00 | 97.50 | 98.50 | 97.00 | 98.00 | 98.50 | 33.00 | 98.0(35%) |
| PassageRetrieval-zh | 96.50 | 96.50 | 51.50 | 76.00 | 89.75 | 95.50 | 92.00 | **97.00** | 96.50 | 17.50 | 96.0(35%) |
| LCC | 53.02 | 53.22 | 52.94 | 54.40 | 54.50 | 54.14 | 53.79 | **54.63** | 53.78 | 52.04 | 53.66(56%) |
| RepoBench-P | 56.83 | 55.89 | 55.47 | **57.55** | 56.76 | 57.05 | 55.68 | 56.38 | 56.36 | 54.67 | 57.38(52%) |

Table 6: LongBench Results with 50% Budget Size on Mistral-7B. .

# H Ablation Study

## H.1 Sparse Regularization Term $\lambda$

We first study the impact of sparse regularization terms $\lambda$. This value controls the efficient KV values cached in the model and thus the model performance. The result is shown in Table 7. We underline the results that are better than the optimal baselines with 25% budgets, and bold the results that are better than the optimal baselines with 50%. Despite that NAtS in Table 2 ($NAtS\ 3e-7$) used more than 25% overall KV budgets for some tasks, here we show $NAtS$ could still outperform many of the corresponding optimal baselines with an even smaller KV budget.

## H.2 Sliding Window Length *W*

Another important hyperparameter for NAtS is the sliding window size *W*. We apply different sliding window sizes *W* ($64, 128, 256, 512$) to fine-tune the Llama3-8B model (with $\lambda = 3e-7$).

The result is shown in Table 8. Overall, all the approaches perform similarly. However, a smaller sliding window size generally results in an overall larger KV cache size. A reduced sliding window size would force the model to apply more Global Tokens and Local Tokens to construct the mid-range

| | Full | NAtS 1e-6 | NAtS 5e-7 | NAtS 3e-7 | NAtS 1e-7 | NAtS 5e-8 |
|---|---|---|---|---|---|---|
| NarrativeQA | 31.4 | 27.05(5%) | 29.19(9%) | 30.53(13%) | 31.39(29%) | 31.94(41%) |
| Qasper | 24.7 | **31.39**(15%) | **28.6**(21%) | **23.76**(27%) | **25.17**(45%) | **26.97**(58%) |
| MultiFieldQA-en | 29.5 | **26.84**(13%) | **27.41**(18%) | **28.94**(24%) | **28.94**(41%) | **29.7**(54%) |
| MultiFieldQA-zh | 60.0 | 51.73(14%) | 56.64(17%) | **61.18**(22%) | **61.27**(37%) | **61.87**(49%) |
| HotpotQA | 17.1 | 15.47(10%) | 17.44(15%) | 16.06(20%) | 17.28(37%) | 16.65(49%) |
| 2WikiQA | 16.6 | 15.37(13%) | 16.44(18%) | **17.05**(23%) | 16.84(41%) | 16.75(53%) |
| Musique | 11.6 | 10.78(9%) | 9.87(13%) | 11.42(18%) | 11.89(36%) | 11.31(48%) |
| DuReader (zh) | 35.6 | 30.06(9%) | **33.63**(13%) | **34.28**(17%) | **33.36**(32%) | **34.41**(45%) |
| GovReport | 34.3 | 30.82(10%) | 32.47(14%) | **33.82**(19%) | **34.23**(36%) | **34.8**(50%) |
| QMSum | 23.3 | 23.37(8%) | 23.09(12%) | 23.11(17%) | 22.6(33%) | 22.74(45%) |
| MultiNews | 27.1 | **26.45**(22%) | **26.57**(28%) | **26.72**(33%) | **27.17**(51%) | **27.11**(64%) |
| VCSUM (zh) | 16.4 | 15.56(7%) | 15.61(10%) | 15.61(14%) | **16.24**(28%) | 15.99(39%) |
| TREC | 72.5 | 68.0(14%) | 70.5(20%) | 72.0(26%) | **73.5**(44%) | **73.0**(56%) |
| TriviaQA | 91.2 | 91.38(12%) | 90.32(17%) | 91.61(22%) | **92.14**(40%) | 91.64(52%) |
| SAMSum | 43.7 | 43.98(8%) | 44.0(12%) | 44.21(16%) | 42.61(30%) | 43.84(41%) |
| LSHT | 46.5 | 46.0(8%) | 44.0(12%) | **47.5**(16%) | 45.5(31%) | 45.5(43%) |
| Passage Count | 6.6 | 2.56(10%) | 6.88(14%) | 7.12(20%) | **9.78**(37%) | **8.82**(50%) |
| PassageRetrieval-en | 98.0 | 48.17(9%) | 82.65(14%) | 95.81(19%) | 97.32(36%) | 96.93(48%) |
| PassageRetrieval-zh | 78.0 | 33.14(12%) | 64.77(16%) | 78.5(20%) | 78.17(34%) | 77.7(46%) |
| LCC | 54.1 | 55.46(24%) | 54.53(31%) | 53.54(38%) | 53.27(57%) | 54.4(69%) |
| RepoBench-P | 51.4 | 50.78(14%) | 51.85(21%) | 53.48(28%) | 52.68(47%) | 52.36(59%) |

Table 7: Ablation Study of Sparse Regularization values $\lambda$ for LLama3.1-8B

| | Full | NAtS 64 | NAtS 128 | NAtS 256 | NAtS 512 |
|---|---|---|---|---|---|
| NarrativeQA | 31.4 | 21.61(33%) | 30.76(15%) | 30.53(13%) | 28.96(12%) |
| Qasper | 24.7 | 17.92(34%) | 24.1(28%) | 23.76(27%) | 28.24(29%) |
| MultiFieldQA-en | 29.5 | 19.37(34%) | 29.02(26%) | 28.94(24%) | 25.87(25%) |
| MultiFieldQA-zh | 60.0 | 35.49(35%) | 60.37(22%) | 61.18(22%) | 58.08(26%) |
| HotpotQA | 17.1 | 13.99(33%) | 16.41(23%) | 16.06(20%) | 17.44(18%) |
| 2WikiQA | 16.6 | 11.52(34%) | 16.69(26%) | 17.05(23%) | 16.48(24%) |
| Musique | 11.6 | 9.42(33%) | 11.62(22%) | 11.42(18%) | 10.89(16%) |
| DuReader (zh) | 35.6 | 22.26(34%) | 35.75(19%) | 34.28(17%) | 33.91(16%) |
| GovReport | 34.3 | 26.63(34%) | 33.89(22%) | 33.82(19%) | 33.05(19%) |
| QMSum | 23.3 | 20.46(33%) | 23.52(19%) | 23.11(17%) | 22.66(16%) |
| MultiNews | 27.1 | 23.78(36%) | 26.73(33%) | 26.72(33%) | 26.46(40%) |
| VCSUM (zh) | 16.4 | 13.8(34%) | 15.61(14%) | 15.61(14%) | 15.9(15%) |
| TREC | 72.5 | 46.5(34%) | 73.0(31%) | 72.0(26%) | 70.5(25%) |
| TriviaQA | 91.2 | 84.44(34%) | 91.89(26%) | 91.61(22%) | 91.66(21%) |
| SAMSum | 43.7 | 38.97(34%) | 43.62(18%) | 44.21(16%) | 44.47(17%) |
| LSHT | 46.5 | 26.5(34%) | 47.0(18%) | 47.5(16%) | 45.5(15%) |
| Passage Count | 6.6 | 0.0(34%) | 5.75(23%) | 7.12(20%) | 5.66(18%) |
| PassageRetrieval-en | 98.0 | 7.13(33%) | 96.71(23%) | 95.81(19%) | 85.78(17%) |
| PassageRetrieval-zh | 78.0 | 5.15(34%) | 81.64(21%) | 78.5(20%) | 70.66(21%) |
| LCC | 54.1 | 39.13(36%) | 54.92(39%) | 53.54(38%) | 54.15(44%) |
| RepoBench-P | 51.4 | 36.87(34%) | 51.89(32%) | 53.48(28%) | 51.34(26%) |

Table 8: Ablation Study of Sliding Window Length $W$ for Llama3

correlation since this distance cannot be covered by the sliding window tokens. However, as the number of sliding window sizes further increases, this compression rate might saturate since the remaining tokens might always require a long-range correlation whose distance is much larger than the sliding window size (e.g., these tokens might require the correlation between two tokens whose distances are larger than 1k or even more).

# I KV Size Distributions

Here, we provide more KV distribution results with different datasets and hyperparameters on the longbench dataset.

Figures 6 and 7 show the KV cache size distributions for Llama 3.1 8B on the Narrative QA and Multi News Datasets. The models tend to preserve more KV caches in the shallower layers and ignore the KV caches in the intermediate layers. However, the model still preserves several KV caches near the output layers. Additionally, even in the same layer, the distributions of the KV cache sizes are not

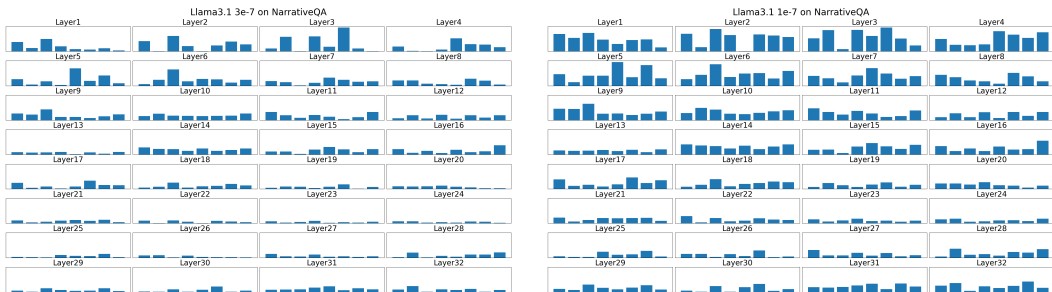

Figure 6: KV size of LLama on Narrative QA Dataset

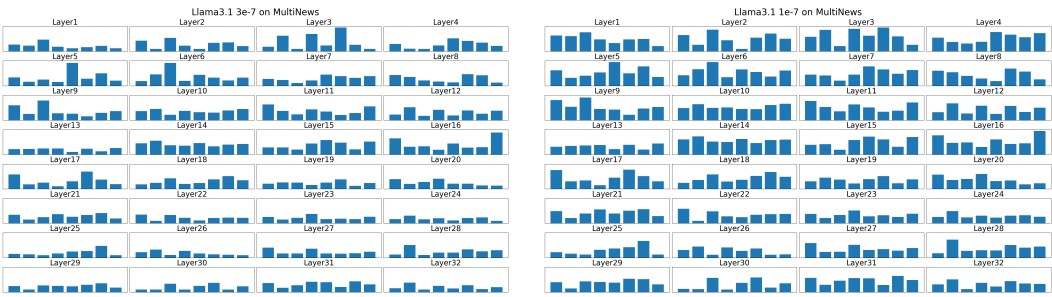

Figure 7: KV size of LLama on Multi-News Dataset

evenly distributed in each layer. Some heads are always preferred while the others might be dropped as the context or $\lambda$ changes.

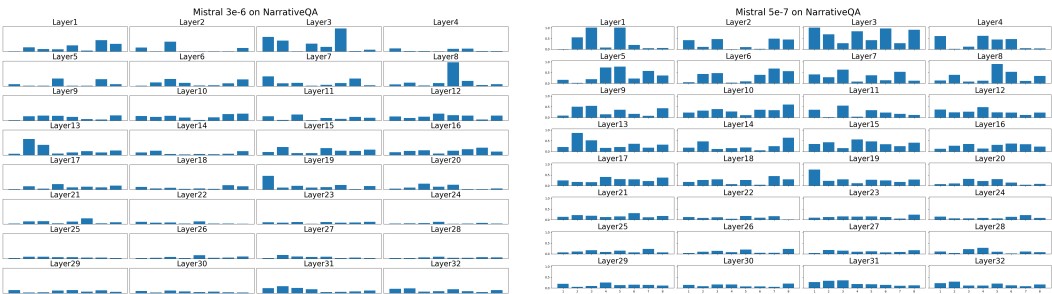

Figure 8: KV size of Mistral on NarrativeQA Dataset

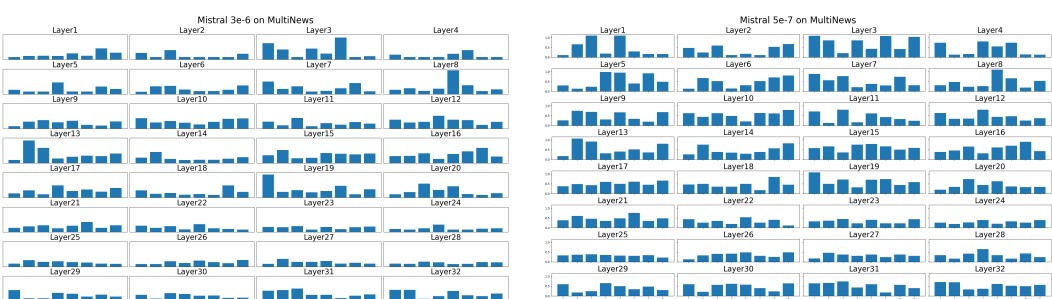

Figure 9: KV size of Mistral on Multi-News Dataset

Figures 8 and 9 show the KV cache size distributions from the Mistral model. Although the two models share some similar trends, e.g., both models tend to preserve more tokens in the shallower and the last few layers. However, compared to Llama, the KV caches for the Mistral model are more unevenly distributed and tend to gather towards some specific heads. Despite that, both Mistral models and LLama models are GQA models [8] and share similar architectures, the preserved token distributions can still be different. This shows that the KV importance distributions might not only depend on the architectures, but are also closely related to the model weights. This highlights the importance of adapting different KV eviction strategies to different models.

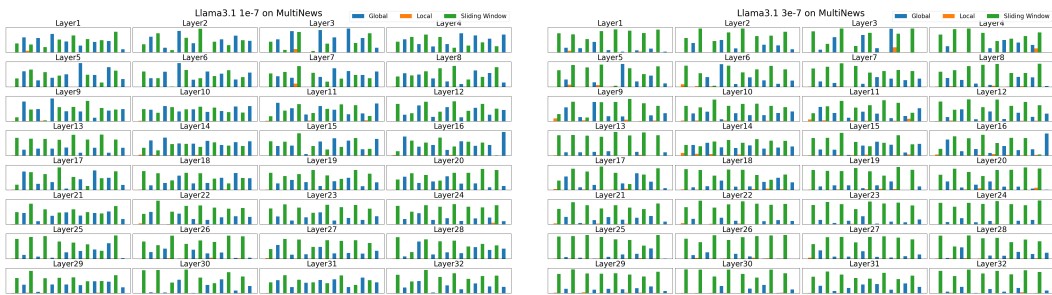

Figure 10: Number of Token Types for Multi-News Dataset

## J  Token Roles Distributions

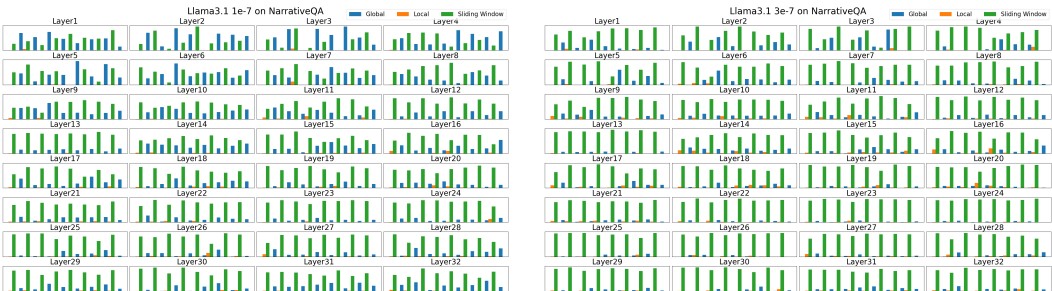

Figure 11: Number of Token Types for NarrativeQA Dataset

We also illustrate the token roles distribution in Figure 10 and 11for the Multi-News and NarrativeQA datasets. Generally, most tokens are classified as either  Global Tokens or  Sliding Window Tokens, and only a few are considered  Local Tokens. This might indicate that most attention operations still focus on either long-range or short-term correlations.

