# OpenReview forum: "Neural Attention Search"
_NeurIPS.cc/2025/Conference — NeurIPS 2025 poster_

### Official Review · Reviewer_UM18 · 2025-06-30

**Clarity:** 3
**Significance:** 3
**Originality:** 2
**Rating:** 5
**Confidence:** 3

**Summary:**

throughout), Local Tokens (survive until the next global token), and Sliding Window Tokens (impact fixed-size windows). Using a Gumbel-Softmax trick and learnable attention masks, NAtS jointly optimizes token type assignments with model weights. The authors demonstrate NAtS's effectiveness through experiments on both training transformers from scratch and fine-tuning existing LLMs, showing significant KV cache reduction while maintaining performance.

**Questions:**

See weaknesses above.

**Ethical Concerns:**

["NO or VERY MINOR ethics concerns only"]

**Final Justification:**

The rebuttal fixed all my concerns, so I raised my score from 4 to 5.

**Limitations:**

See weaknesses above.

**Quality:**

3

**Strengths And Weaknesses:**

### Strengths
- Well-structured presentation. The paper is quite readable with clear explanations of the approach.
- Strong empirical results. The experiments are thorough. They test on multiple benchmarks (RULER, LongBench), different models (LLama 3.1, Mistral-7B), various KV budget levels, and compare against many baselines (SnapKV, AdaKV, PyramidKV). NAtS consistently outperforms baselines.
- Practical implementation details provided. The authors integrate their approach into FlashAttention (Algorithms 1-2 in the appendix), showing how to efficiently implement the sparse attention patterns. They also provide concrete latency/memory measurements in Figure 4.

### Weaknesses
- Limited novelty in core concepts. The idea of having different token types (global, local, sliding window) isn't particularly new - the paper itself mentions these patterns exist in StreamingLLM, Longformer, etc. The main contribution seems to be learning which tokens get which role, but this feels somewhat incremental.
- Efficiency gains appear limited to very long contexts. Looking at Figure 4, the memory/latency benefits only become significant when context length exceeds 100k tokens. For more typical contexts (under 50k), the improvements seem marginal. This limits the practical impact since most real-world applications don't use such extreme context lengths.
- Training overhead was not clearly discussed. While inference efficiency is well-studied, the paper doesn't clearly quantify the overhead of learning these token assignments during training. The additional Attention Score Layer parameters and Gumbel-Softmax sampling likely add computational cost, but this isn't analyzed.

---

> ### Author Rebuttal · Authors · 2025-07-30
>
> Dear reviewer UM18,
>
> we thank you for your positive feedback regarding the efficiency of NAtS, regarding your concerns:
>
> > Limited novelty in core concepts. The idea of having different token types (global, local, sliding window) isn't particularly new - the paper itself mentions these patterns exist in StreamingLLM, Longformer, etc. The main contribution seems to be learning which tokens get which role, but this feels somewhat incremental.
>
> Although the token types are defined in previous works, finding the optimal token types for each token is still prohibitive, as the potential search space contains $3^{L×H ×N}$ options. Where $L, H, N$ are the sequence length, number of heads, and number of layers in the network. For a Llama model with 8 KV heads, 32 layers, even checking the type of one single token results in a complexity of $3^{256} \approx 1.39\times 10^{122}$ options. This value can even grow with the increased number of observed tokens. The predefined rules can only cover a tiny fraction of this search space.
>
> Additionally, despite some other works such as SeerAttention[1] and Tokenbutler[2] also trying to learn a sparse attention map, they require a pre-trained model as a teacher model and cannot be learned end-to-end. In contrast, NAtS learns its token types directly from the gradient information, using either next-token prediction loss without any predefined rules.
>
> In this paper, we show that:
>
> 1. A learnable attention mask can be learned jointly with the transformer parameters towards a sparser transformer.
>
> 2. The gradient of attention masks can be gathered to indicate the importance of each token and backpropagated towards their types.
>
> 3. In the end, we achieve a token-level sparsity control that best trades off sparsity vs. model performance.
>
> For a NAS problem [3], the search strategies are important as the search space design; hence, knowing how to search for the optimal token types should not only be considered incremental compared to the search space construction.
>
>
> > Efficiency gains appear limited to very long contexts. Looking at Figure 4, the memory/latency benefits only become significant when context length exceeds 100k tokens. For more typical contexts (under 50k), the improvements seem marginal. This limits the practical impact since most real-world applications don't use such extreme context lengths.
>
> The efficiency of LLM inference is determined by many factors; for instance, the memory used to store the model parameters is much larger than the KV cache if the context length is not large enough. The cost of feedward layers cannot be negligible when the input context length is small. While for the decoding stage, memory bounds could be a bottleneck with the shorter content length.  Here, we mainly focus on the cases where the computational bounds are the bottleneck to highlight the efficiency of NAtS.
>
> Here, we also show the efficiency gains for the pre-filling (computationally dense) stages with a prefilling length smaller than 100k tokens. With a sparse kernel designed for first computing attention value for the global/local and then the values for the sliding window tokens, NAtS provides an overall $2.35 \times$ lower latency compared to vanilla transformer within token budgets smaller than 100k
>
> |      |   10k |   20k |   30k |   40k |   50k |   60k |   70k |   80k |   90k |   100k |
> |:-----|--------:|--------:|--------:|--------:|--------:|--------:|--------:|--------:|--------:|---------:|
> | FA2  |    0.69 |    1.53 |    2.51 |    3.75 |    5.24 |    6.9  |    8.91 |   10.97 |   13.44 |    16.43 |
> | NAtS |    0.59 |    1.13 |    1.71 |    2.37 |    3.08 |    3.76 |    4.55 |    5.29 |    6.16 |     6.98 |
>
>
> > Training overhead was not clearly discussed. While inference efficiency is well-studied, the paper doesn't clearly quantify the overhead of learning these token assignments during training. The additional Attention Score Layer parameters and Gumbel-Softmax sampling likely add computational cost, but this isn't analyzed.
>
> The Attention score layer is a linear layer that maps the input feature $X \in R^{L \times D}$ to $Logits~\in R^{L \ times (n_{ops}*h_{kv})}$ where $n_{opts}$ and $h_{kv}$ are the number of token types and number of KV heads respectively. This cost is minimal compared to the costs of computing the QKV values (roughly $n_{ops}/d_{head}$) where $d_{head}$ is the dimension of each head. Hence, overall, this operation also introduces a complexity of $O(L)$. The Gumbel-Softmax sampling only requires the token-level softmax and argmax operations across different types and thus does not introduce a complexity of $O(L)$ and is theoretically smaller than the overhead from the linear layers. The additional training overhead that NAtS brings to vanilla transformer for the pre-training tasks is only roughly 5% (0.93s per iteration for NAtS vs. 0.88s per iteration for vanilla transformer)
>
>
> [1] Gao et al. SeerAttention: Learning Intrinsic Sparse Attention in Your LLMs
>
> [2] Akhauri et al. TokenButler: Token Importance is Predictable
>
> [3] Elsken et al. Neural architecture search: A survey.

---

> > ### Comment · Reviewer_UM18 · 2025-08-06
> >
> > Thank you for the authors' response. It fixed all three of my concerns about the weakness. Thus, I will increase my score from 4 to 5.

---

### Official Review · Reviewer_6LT7 · 2025-07-03

**Clarity:** 4
**Significance:** 4
**Originality:** 4
**Rating:** 4
**Confidence:** 4

**Summary:**

This paper proposes a neural attention search (NAtS) for three token types: global tokens, local tokens, and sliding window tokens. By learning an attention mask, NAtS is able to jointly learn token type information with architecture weights, similar to the single-shot neural architecture search method. The authors demonstrated in experiments training Transformer from scratch and fine-tuning existing large language models (LLMs) that NAtS can effectively reduce the KV cache size and reduce inference cost while maintaining model performance.

**Questions:**

NAtS can automatically learn token types. How does the ratio of global, local, and sliding window tokens change under different tasks and context lengths? Is this ratio change related to task characteristics (e.g., some tasks require stronger global understanding, while others focus more on local information)?

NAtS treats token type search as a NAS problem. Compared with typical NAS methods, what are the characteristics of NAtS in terms of computational complexity or training stability during the search process? How does this "optimizing token state information and model weights in a single forward and backward propagation" approach reduce the computational overhead of NAS?

Is there a more theoretical or in-depth analysis to explain how exactly λ guides the model to learn a sparse attention map? In practical applications, how do we determine the best λ value to strike a balance between performance and efficiency? Can we dynamically adjust λ based on the task or context?

Figures 8 and 9 show the distribution of KV cache sizes across layers and heads, and mention that the KV cache distribution in the Mistral model is more uneven than that in Llama. Is this unevenness a feature of the Mistral model design itself (such as GQA/MQA), or does it reflect the specific sparsity patterns learned by NAtS on different model architectures? What guidance do these observations have for future research on sparse attention?

**Ethical Concerns:**

["NO or VERY MINOR ethics concerns only"]

**Final Justification:**

Thank you for addressing some of my concerns. However, the comparison between methods that require training and those that do not still lacks fairness. I believe in and appreciate the potential and contributions of this work, but the authors may need to conduct more equitable experiments. Therefore, I have raised my scores for Clarity, Significance, and Originality.

**Limitations:**

yes

**Paper Formatting Concerns:**

Some tables may have formatting issues (too long or not standardized)

**Quality:**

3

**Strengths And Weaknesses:**

+ The most notable advantage of NAtS is its ability to learn token importance end-to-end. By treating token role assignment as an optimization problem, NAtS leverages the principles of neural architecture search to jointly optimize token roles and model weights, avoiding the limitations of hard-coded patterns or heuristic rules commonly found in traditional sparse attention methods.
+ NAtS can effectively reduce memory usage and accelerate reasoning in both the pre-filling and decoding stages, improving the efficiency and scalability of long text reasoning.
+ NAtS is integrated into FlashAttention 2, which helps avoid explicit calculation of attention masks during forward propagation and can skip calculation blocks that do not contain valid values, further improving efficiency.
- Although the search space of NAtS is flexible, the paper acknowledges that the construction of the attention mask is still column-oriented and focuses mainly on vertical-style sparse attention. Other important sparse attention types such as block-sparse attention and slash attention are not involved.
- Compared with other training-free methods, NAtS requires training, which makes it difficult to apply it to large-scale models.
- The paper mentions the time and number of GPUs required to train the Transformer model and fine-tune the LLM in NAtS, but there is a lack of in-depth analysis on the training complexity of the "Attention Score Layer" and whether this "joint optimization" method has significant advantages in training time or resource consumption compared to traditional NAS methods.
- Although the paper conducted an ablation study of λ and W, a theoretical explanation or a more fine-grained empirical analysis of how λ precisely controls sparsity and its impact on model performance and KV cache size can be more in-depth. For example, why can the KV cache still be reduced to 35% when λ is set to 0? The impact of the sliding window size W on performance and sparsity is also worth further exploring its internal mechanism.
- The appendix shows the distribution of KV cache sizes at different layers and heads. The reasons behind these distribution patterns or their deep connection with model performance and task types can be explained in more depth.

---

> ### Author Rebuttal · Authors · 2025-07-30
>
> Dear reviewer 6LT7,
>
> We thank you for highlighting the flexibility and the efficiency of NAtS, regarding your concerns:
>
> > Other important sparse attention types such as block-sparse attention and slash attention are not involved.
>
> We agree that NAtS only incorporates the column-wise attention patterns. However, this decision also helps to efficiently reduce the KV cache size. The block-sparse and slash patterns must maintain the entire KV caches, as the valid values in the block-sparse and slash patterns need to be determined by the joint values of the Q, K, and V. Hence, we cannot easily discard KV caches, as they might be required by the following Q. Additionally, including the block-sparse and slash patterns might require additional checks for the types of the Q values, increasing the complexity during both training and testing phases from $O(L)$ to $O(L^2)$ as we need to check the types of both Q and KV values, which might be pragmatic with the increased sequence length.
>
> > NAtS requires training, which makes it difficult to apply it to large-scale models.
>
> As NAtS only trains the projection layers, which only provides a reduced memory overhead compared to fine-tuning a full model. Hence, we only need 2 H100 GPUs to fine tune an 8B LLM.
>
> Among the existing training-free methods, H2O requires the explicit attention map values, which might requires lots of GPU memories and is not compatible with flash attention; Streaming LLM cannot get information in the middle of the context; SnapKV based approaches select the KV values based on the last Q values and thereby, relies heavily on the positions of Q values, which might lose essential information for scenarios like multi-round diagonal or reasoning models. Additionally, the pre-defined rules (such as the uniformly distributed token budgets for SnapKV, layerwise gradually decreased KV budgets for PyramidKV, or layerwise uniformly budget distributions for AdaKV) might not generalize to all the architectures and contexts.
>
> Hence, since all these training-free approaches only make predictions based on the past information, they might fail to preserve the information that might be required by a longer context such as reasoning models or multi-round dialogue scenarios.
>
> > but there is a lack of in-depth analysis on the training complexity of the "Attention Score Layer" and whether this "joint optimization" method has significant advantages in training time or resource consumption compared to traditional NAS methods.
>
> The Attention score layer is a linear layer that maps the input feature $X \in R^{L \times D}$ to $Logits~\in R^{L \ times (n_{ops}*h_{kv})}$ where $n_{opts}$ and $h_{kv}$ are the number of token types and number of KV heads respectively. This cost is minimal compared to the costs of computing the QKV values (roughly $n_{ops}/d_{head}$) where $d_{head}$ is the dimension of each head. Hence, overall, this operation also introduces a complexity of $O(L)$.
>
>  The traditional NAS approach approximates a bilevel optimization process by updating the model weights and architecture parameters alternatively. Hence, within each updating iteration, only a small fraction of weights are updated. Since the number of parameters in the projection layer is much smaller than the number of parameters in the network, this might not be data-efficient. Hence, we propose to optimize the projection layers and the model parameters jointly to make full use of the training data, which is 2 times more data efficient than the bilevel optimization approaches in NAS frameworks. This approach is also applied in the mixture-of-expert models to optimize the gating parameters and model parameters jointly, showing its efficiency in training larger- scale models.
>
> > a theoretical explanation or a more fine-grained empirical analysis of how λ precisely controls sparsity and its impact on model performance and KV cache size can be more in-depth.
>
> The $\lambda$ value is directly applied to the gradients of each logit for the token types. Hence, it can be considered as a regularization term that encourages more sliding window tokens.
>
> More specifically, the gradients for each logit value are the column-wise sum of the corresponding attention mask values. As discussed in Appendix C, the gradients for the attention masks are the same as the gradients for each attention map value if no masks are applied (except for the normalization denominator). Hence, if the column-wise sum of the gradients for each iteration is smaller than $\lambda$, indicating that the attention maps of that column requires no further update or gradients updates towards a sparser transformer, they will be push the projection layer to classify those tokens as sliding window tokens.  In contrast, if the column-wise tokens actually require larger attention map values, their corresponding gradients will be larger than $\lambda$ and push the model towards global/local tokens. This ensures that the global tokens and local tokens actually require a certain amount of attention map values to keep them activated. Hence, this value could also be considered as a soft threshold for the attention map values, where attention map values smaller than that will be encouraged to be filtered out.
>
> The reason that KV cache reduced to 35% when $\lambda$ is set to 0 might be the result of the combination of Global Tokens and Local tokens, where local tokens encourage the correlations for the points that are closer to each other. Thus, we can drop those local tokens earlier if they might increase the model loss if those corresponding attention map values increase (as shown in Section C.2 and Figure 5). Additionally, some of the attention heads might only focus on the local dependencies. These heads might naturally surpass the attention map values with long-range correlations and therefore would prefer the sliding window tokens.
>
>  We briefly discuss the impact of W and sparsity in Section G.2. Additionally, since the LLMA models use rope as their positional encoder, while the rope values decay as the distance between two data points increases, the sliding window size vs. sparsity might also be correlated to the rope parameters.
>
>
> > The appendix shows the distribution of KV cache sizes at different layers and heads. The reasons behind these distribution patterns or their deep connection with model performance and task types can be explained in more depth.
>
> NAtS learns the tokens roles only with the gradient information and without any prior information.  Overall, the distributions fit the observation from other works, such as the higher and intermediate layers are less important than the first few initial layers or the output layers. Additionally, only a partial fraction of heads are responsible for constructing long-term distributions, which are well supported by many architecture pruning works. However, as motivated in the paper, we are not trying to find a fixed pattern that works for any specific architectures or tasks, but are trying to ask the models to learn this information by themselves.
>
>
> > How does the ratio of global, local, and sliding window tokens change under different tasks and context lengths?
>
> Overall, the fraction of tokens changes across different tasks. For instance, on a RepoBench-P dataset, the fraction of global, local, and sliding window tokens is 23\%, 2\%, 75\%, respectively. In contrast, for longer QA tasks, such as NarrativeQA, the fractions become 11\%, 3\%, 85\%, respectively. Since the information density of codes is much larger than the information contained in natural languages. This further shows the importance of data-dependent budget allocation for different tasks
>
> > How does this "optimizing token state information and model weights in a single forward and backward propagation" approach reduce the computational overhead of NAS?
>
> Please check our replies for weakness “whether this "joint optimization" method has significant advantages in training time or resource consumption compared to traditional NAS methods.”
>
> > Is there a more theoretical or in-depth analysis to explain how exactly λ guides the model to learn a sparse attention map?
>
> For how $\lambda$ influences the sparsity, please refer to our replies to “a theoretical explanation or a more fine-grained empirical analysis of how λ precisely controls sparsity and its impact on model performance and KV cache size can be more in-depth”
>
> Regarding the best $\lambda$ values, since the sparsity values can already be observed in the early training stage (Line 274), we could determine if we would like to early-stop runs that cannot achieve the desired sparsity. Alternatively, we could also construct different $\lambda$ values as different experts and consider the entire system as a mixture-of-expert system to propose the optimal $\lambda$ values for the target tasks. Since $\lambda$ controls the sparsity by directly modifying the weights for the token type logits, we could also control the $\lambda$ values with the gradient information from the attention output $dO$, such as its norm values. However, for the sake of simplicity, we set a fixed value in our experiment.
>
> > Is this unevenness a feature of the Mistral model design itself (such as GQA/MQA), or does it reflect the specific sparsity patterns learned by NAtS on different model architectures? What guidance do these observations have for future research on sparse attention?
>
> Both Mistral and LLama3 models are GQA models. Hence, this might be due to the different distribution of the two models parameters. For instance, most heads of mistral models might only focus on the local information with a sliding window: the attention maps of points that are far away to each other might not be large enough and can be removed; only a few heads are responsible for the global information. In LLama, more heads might contribute to the final predictions and thus we need to preserve the tokens from those heads.

---

> > ### Comment · Reviewer_6LT7 · 2025-08-05
> >
> > Thanks to the author for the detailed reply, I have improved my score.

---

> > > ### Author Response · Authors · 2025-08-05
> > > **Thanks!**
> > >
> > > Many thanks for improving your score!
> > > Surprisingly, we don't see your rating anymore. Do you know whether this is an intentional setup of OR?

---

### Official Review · Reviewer_GRci · 2025-07-03

**Clarity:** 2
**Significance:** 3
**Originality:** 3
**Rating:** 4
**Confidence:** 2

**Summary:**

The paper proposes Neural Attention Search (NAtS), a learnable sparse attention framework that assigns each token a discrete role—Global, Local, or Sliding Window—to control its lifespan and influence in attention computation. By jointly optimizing these token roles and model weights via a differentiable mechanism inspired by neural architecture search (NAS), the method aims to reduce KV cache size and inference cost, while maintaining the model's performance on long-context tasks.

**Questions:**

- How do you envision your method being applied to reasoning tasks that generate long sequences?

- Can the framework be extended to support adaptive role reassignment during generation, or is it fixed once the role is predicted? What about for those multi-step tasks?

**Ethical Concerns:**

["NO or VERY MINOR ethics concerns only"]

**Final Justification:**

I find the authors’ explanations regarding my concerns persuasive, and I have revised my score to 'weak acceptance.'

**Limitations:**

yes

**Quality:**

2

**Strengths And Weaknesses:**

Strengths

- The idea of casting sparse attention as a NAS problem is novel and provides a structured lens through which to learn token importance.

- The method achieves both memory and latency reductions, which are crucial in long-context LLM inference and deployment.

Weaknesses

- While the NAS formulation is interesting, the underlying insight and motivation for using the NAS method is somewhat unclear. For example, other learning-based methods like SeerAttention directly learn the sparse attention mask—what are the practical trade-offs between learning token roles vs. learning masks? More discussion would help clarify the positioning.

- The search space (Global / Local / Sliding Window tokens) feels hand-designed. It is hard to judge the validity of this space. For example, why are Local tokens dropped after a Global token? It would be helpful to explore or justify alternative or more flexible formulations.

- The latency and memory results only compare with full attention, not with other sparse attention baselines (e.g., SeerAttention, MInference, DuoAttention). This makes it hard to judge the relative efficiency gains.

- Section 3.3 (Efficient Inference) would benefit from a clearer and more formal algorithm description or pseudocode. The current writing is hard to follow and lacks clarity on KV cache update logic across heads and layers.

---

> ### Author Rebuttal · Authors · 2025-07-30
>
> Dear reviewer GRci,
>
> We thank you for agreeing with  NAtS’s efficiency and novelty. Regarding your concerns:
>
>
> > While the NAS formulation is interesting, the underlying insight and motivation for using the NAS method is somewhat unclear. For example, other learning-based methods like SeerAttention directly learn the sparse attention mask—what are the practical trade-offs between learning token roles vs. learning masks? More discussion would help clarify the positioning.
>
>
> Both NAtS and SeerAttention propose learned sparse attention maps instead of fixed attention patterns. However, SeerAttention is trained to approximate the attention output of each layer from an existing dense transformer that only involves the QK matrix, whereas NAtS learns the token roles directly from the final loss function in an end-to-end manner that involves the QKV matrices. Hence, SeerAttention, as a post-hoc process, requires a dense transformer as a teacher model and can only be applied during the post-training stage. In contrast, NAtS does not necessarily require the teacher model and can be learned from scratch, allowing it to be used in both pre-training and post-training stages.
>
> Additionally, SeerAttention still needs to maintain the full KV caches, as the decision of whether a block should be computed is determined by the gating values from both Q and K. There is no information telling the optimizer when a KV cache can be dropped. In contrast, NAtS evicts the tokens on the fly once the corresponding KV values are no longer required in the following attention computation process, and thus could efficiently reduce the GPU memory consumption.
>
> Last but not least, SeerAttention is only applied during the pre-filling stage, while NAtS can be used in both the pre-filling and decoding stages, as NAtS checks the types of each newly generated token and evicts non-global tokens with the corresponding rules for each token (as shown in Section 3.3) and thus could further reduce the computational costs for models with long outputs such as reasoning models.
>
> Thank you for pointing this out. We will also describe the differences between our work and the existing sparse attention approach in our paper.
>
>
>
> > The search space (Global / Local / Sliding Window tokens) feels hand-designed. It is hard to judge the validity of this space. For example, why are Local tokens dropped after a Global token? It would be helpful to explore or justify alternative or more flexible formulations.
>
> Global tokens are the default settings in transformer models, while sliding window tokens are widely used in other sparse transformer models [1,2]. Therefore, we include them in the NAtS search space. These two token types can be considered as two extremes: global tokens will be kept forever, while sliding window tokens will only be kept for a fixed number of steps. However, in practice, we might also face cases where tokens need to be preserved longer than the existing sliding window size, but are no longer required after that. For instance, a piece of code might be important in its package but might not be necessary when its functions are imported into other packages. Since we cannot expect this distance to remain constant, as it depends on contextual information, we introduce local tokens whose lifetimes are learned during training. Hence, local tokens can be considered as a sliding window token with learnable sliding window sizes. Since this token type is an intermediate between global and sliding window tokens, we apply a global token as a signal that ends the lifetime of the local tokens. As we primarily focus on column-wise sparse attention, an alternative would be the introduction of sliding window tokens from different sliding window shapes, which could be covered by the local tokens.  We believe that these three roles are already sufficiently flexible to cover a wide range of different attention maps.
>
>
>
> > The latency and memory results only compare with full attention, not with other sparse attention baselines (e.g., SeerAttention, MInference, DuoAttention). This makes it hard to judge the relative efficiency gains.
>
> Since the other sparse attention baselines (SeerAttention, MInference) only work for the pre-filling stages and do not evict the KV cache, we mainly focus on the pre-filling latency here. To show the efficiency of NAtS during the pre-filling stage, we conduct experiments comparing the latency of NAtS against other baselines. With a sparse kernel designed for first computing attention value for the global/local and then the values for the sliding window tokens,  NAtS provides an overall lower latency compared to Duo Attention and Seer Attention across all context lengths, and better or comparable to MInference until 150k Context length. However, MInference can only be applied to accelerate the pre-filling stage and cannot provide further benefits towards memory reduction or decoding speed up, and thus cannot scale to a higher context length due to the GPU memory limits.
>
> | Prefilling Size |   FA2 |   MInference |   DuoAttn |   SeerAttn |   NAtS |
> |-------:|------:|-------------:|----------:|-----------:|-------:|
> |  10k |  0.69 |         1.53 |      1.11 |       0.48 |   0.59 |
> |  50k |  5.24 |         4.8  |      3.32 |       3.14 |   3.08 |
> | 100k | 16.43 |         7.99 |      8.03 |       8.14 |   6.98 |
> | 150k | 33.21 |        11.51 |     14.59 |      14.91 |  12.41 |
>  | 200k | 56.98 |        15.74 |     22.1  |      24.15 |  19.22 |
>
>
>
>
> > Section 3.3 (Efficient Inference) would benefit from a clearer and more formal algorithm description or pseudocode. The current writing is hard to follow and lacks clarity on KV cache update logic across heads and layers.
>
> Thanks for pointing this out. We will add pseudo-code to describe the process more precisely. The main message here is that we
>
> 1. check the types of the KV cache from each head;
>
> 2. Move the most recent sliding window token to the sliding window queues (located at the beginning of the KV caches);
>
> 3. Check if global tokens are contained in the new tokens. If so, remove all local tokens from the corresponding heads and move the global tokens after the last global token of that head, and remove the remaining local tokens of that head.
>
>  Thus, we could efficiently reduce the required KV caches and, furthermore, the corresponding computational times.
>
>
>
> > How do you envision your method being applied to reasoning tasks that generate long sequences?
>
>
> As NAtS dynamically checks the token roles and evicts the local and sliding window tokens, regardless of whether the token belongs to the prefilling stages or is generated by the transformers, NAtS can be directly applied during the decoding stages. Additionally, since the NAtS token roles are independent of the Q values, following the KV cache update algorithm described above, NAtS can be directly applied to reasoning tasks without any further adjustment.
>
>
> > Can the framework be extended to support adaptive role reassignment during generation, or is it fixed once the role is predicted? What about for those multi-step tasks?
>
>
> Once predicted, the token roles will be fixed. However, since we only update the projection layers (i.e., layers that map the input feature map values to the token types) with the next-token-prediction loss and do not introduce any auxiliary information about whether the corresponding token should be preserved, the roles of each token are assigned based on their impacts on all the potential following tokens and therefore should also provide further information in multi-step tasks. Additionally, since the token roles are not dependent on the Q values, we can ensure that the tokens preserved are also ready to be queried in the following steps, as the tokens that might only be used in the current step would be classified as local and sliding window tokens. Hence, despite the fixed token roles, NAtS can still be directly applied to multi-step tasks.
>
> [1] Child et al. Generating long sequences with sparse transformers
>
> [2] Xiao et al. Efficient Streaming Language Models with Attention Sinks

---

> > ### Comment · Reviewer_GRci · 2025-08-06
> >
> > Thank you for the efforts to address my concerns! I will improve my score.

---

> > > ### Author Response · Authors · 2025-08-06
> > > **Thanks!**
> > >
> > > Dear Reviewer GRci,
> > >
> > > Many thanks for the response and for increasing your scores!

---

### Official Review · Reviewer_1gKy · 2025-07-12

**Clarity:** 3
**Significance:** 3
**Originality:** 2
**Rating:** 4
**Confidence:** 3

**Summary:**

Neural Attention Search (NATS) introduces an end-to-end learnable sparse-attention framework that lets a transformer decide, during training, how long each token should remain in the key–-value cache.  Each token is classified as a global, Local, or Sliding-Window token via a Gumbel-Softmax search over a huge architectural space, and the chosen roles are encoded as a learnable attention mask. When applied to GPT-2-small (trained from scratch on PG-19) and to fine-tuned Llama-3-8B / Mistral-7B, NAtS cuts KV-cache size to 3 % – 50 % of full attention with negligible perplexity loss, enabling context windows up to 700 k tokens and 1.8×–2.6× speed-ups on an H100 GPUs

**Questions:**

- Could you report mean ± std over seeds (or another statistical test) for key metrics to substantiate that the observed ≤1 pp performance gaps are robust
- does the per-token Gumbel sampling introduce communication overhead, and could the learned masks generalize across TP shards without retraining

**Ethical Concerns:**

["NO or VERY MINOR ethics concerns only"]

**Final Justification:**

Thanks for the authors for the responses, they largely resolve my concerns. I raised my score accordingly.

**Limitations:**

no negative societal impact

**Quality:**

3

**Strengths And Weaknesses:**

strength:
- the token-type search is optimized jointly with network weights and implemented inside FlashAttention 2, avoiding explicit masks.
- Pushes single-GPU context length from 200 k to 700 k tokens with 2.24× memory savings during pre-fill.
weakness
- The smallest lambda settings still incur up-to-34 % perplexity loss on certain tasks when KV budget falls below ~5%.
- Search space is column-oriented; block-sparse and slash patterns are excluded

---

> ### Author Rebuttal · Authors · 2025-07-30
>
> Dear reviewer 1gKy, we thank you for your constructive comments,
>
> > The smallest lambda settings still incur up-to-34 % perplexity loss on certain tasks when KV budget falls below ~5%.
>
> Thanks for your comment, but we are not quite sure which result you refer to.
>
> If we understand your comments correctly, you are referring to Figure 3 with the pre-training tasks, where the sparest model only requires a KV cache size of 2.3% to achieve a perplexity of 14.38. This perplexity value is only roughly 2.8% higher than the full transformer (13.99). Another candidate that requires 5.69% KV cache size achieves a perplexity of 14.08, only 0.6% higher than the full attention module.
>
> The 34% perplexity loss might come from the highest value in Figure 3 (Streaming LLM with 12.5% KV cache size). In comparison, Streaming LLM receives a 19% perplexity loss with a KV cache size of 12.5% while H2O gets a 12% perplexity loss with KV cache size of  6.25%, both loss-wise and KV cache budget-wise worse than NAtS.
>
> > Search space is column-oriented; block-sparse and slash patterns are excluded
>
> We agree that NAtS only incorporates the column-wise attention patterns. However, this decision also helps to efficiently reduce the KV cache size. The block-sparse and slash patterns must maintain the entire KV caches, as the valid values in the block-sparse and slash patterns need to be determined by the joint values of the Q, K, and V. Hence, we cannot easily discard KV caches, as they might be required by the following Q. Additionally, including the block-sparse and slash patterns might require additional checks for the types of the Q values, increasing the complexity during both training and testing phases from $O(L)$ to $O(L^2)$ as we need to check the types of both Q and KV values.
>
> > Could you report mean ± std over seeds (or another statistical test) for key metrics to substantiate that the observed ≤1 pp performance gaps are robust
>
> Thanks for the advice. We have retrained some of the NAtS models ($\lambda$ values with $0, 1e-8, 5e-9$). Due to the expense of training an LLM from scratch, we cannot afford the cost of training all the models from scratch with multiple seeds. The results are shown as follows:
>
> |             |   KV Budget | Perplexity     |
> |:------------|------------:|:---------------|
> | NAtS        |       0.338 | 13.980 (0.007) |
> | Transformer |       1     | 13.991 (0.007) |
> | H2O         |       0.5   | 14.048 (0.009) |
> | NAtS        |       0.093 | 14.074 (0.016) |
> | NAtS        |       0.057 | 14.104 (0.016) |
> | H2O         |       0.25  | 14.241 (0.012) |
> | H2O         |       0.125 | 14.704 (0.011) |
> | Streaming   |       0.5   | 14.730 (0.150) |
> | Streaming   |       0.25  | 15.431 (0.218) |
> | H2O         |       0.062 | 15.743 (0.027) |
> | Streaming   |       0.125 | 16.375 (0.269) |
> | Streaming   |       0.062 | 17.729 (0.339) |
> | Streaming   |       0.031 | 19.795 (0.444) |
>
> We show the mean and std of the perplexity values with three repetitions in the table above. Despite that NAtS (with $\lambda$ as 0) only requires around 33.8% of the required KV budgets, it achieves a lower perplexity compared to a full transformer with 100% KV budgets. Additionally, the table above is consistent with the results shown in Figure3: NAtS could achieve a much lower mean perplexity within the same budgets compared to the other baselines.
>
>
> > does the per-token Gumbel sampling introduce communication overhead, and could the learned masks generalize across TP shards without retraining
>
> The generated Gumbel samples only have a matrix of size $L \times H \times N_{ops}$ where L, H, and $N_{opts}$ are the sequence length, number of KV heads and number of operations, respectively. This is much smaller than the QKV tensors with size $L \times H \times D_{heads}$ where $D_{heads}$ is the head dimension. Additionally, we could also split head-wise the Gumbel samples (and their corresponding weights) across different GPUs, where each GPU only receives the Gumbel samples from the corresponding heads. The masks are generated for each head dynamically and independently within the computational kernels during both the training and validation process, and therefore could be directly applied across different TP shards without retraining.

---

> > ### Author Response · Authors · 2025-08-08
> >
> > Thanks for being a reviewer for our paper. Since we haven't heard back from you after our reply, we hope that we have resolved all your concerns and answered all your questions.

---

### Decision · Program_Chairs · 2025-09-17

**Decision:**

Accept (poster)

**Comment:**

Neural Attention Search is an end-to-end learnable sparse transformer that automatically evaluates token importance to reduce KV cache and inference costs while maintaining performance. Key strengths include its novel casting of sparse attention as a Neural Architecture Search problem, end-to-end learnability, and integration into FlashAttention 2 for efficiency. It effectively reduces memory and accelerates reasoning in both pre-filling and decoding stages. Empirical results demonstrate strong performance across various benchmarks and models.

Initial weaknesses identified by reviewers involved the hand-designed nature of the search space, a perceived lack of novelty in core concepts, limited comparisons against other sparse attention baselines, and insufficient discussion of training overhead.

During the rebuttal, authors provided convincing responses. They clarified NAtS's minimal perplexity loss compared to baselines and its advantages over other sparse methods in terms of end-to-end learning and full-stage application. They provided additional latency comparisons against other sparse attention techniques and quantified the minimal training overhead. The authors also justified their search space design and the benefits of their joint optimization approach. All reviewers (1gKy, GRci, 6LT7, UM18) acknowledged that their concerns were addressed and consequently increased their scores. Based on the robust technical contributions and effective rebuttal, the paper is recommended for acceptance.